# Severe fever with thrombocytopenia syndrome virus induces lactylation of m6A reader protein YTHDF1 to facilitate viral replication

Bingxin Liu [1], Xiaoyan Tian[1], Linrun Li[1], Rui Zhang[2], Jing Wu [1], Na Jiang [1], Meng Yuan[1], Deyan Chen [1], Airong Su[3], Shijie Xu[1] & Zhiwei Wu [1,4,5]✉

## Abstract

**Severe fever with thrombocytopenia syndrome virus (SFTSV), an emerging infectious pathogen with a high fatality rate, is an enveloped tripartite segmented single-stranded negative-sense RNA virus. SFTSV infection is characterized by suppressed host innate immunity, proinflammatory cytokine storm, failure of B-cell immunity, and robust viral replication. m6A modification has been shown to play a role in viral infections. However, interactions between m6A modification and SFTSV infection remain poorly understood. Through MeRIP-seq, we identify m6A modifications on SFTSV RNA. We show that YTHDF1 can bind to m6A modification sites on SFTSV, decreasing the stability of SFTSV RNA and reducing the translation efficiency of SFTSV proteins. The SFTSV virulence factor NSs increases lactylation of YTHDF1 and YTHDF1 degradation, thus facilitating SFTSV replication. Our findings indicate that the SFTSV protein NSs induce lactylation to inhibit YTHDF1 as a countermeasure to host's YTHDF1-mediated degradation of m6A-marked viral mRNAs.**

**Keywords** SFTSV; m6A; Post-translational Modifications (PTMs); Lactylation; YTHDF1
**Subject Categories** Microbiology, Virology & Host Pathogen Interaction; Post-translational Modifications & Proteolysis; RNA Biology

## Introduction

Tick-borne infectious diseases are a growing threat to public health on a global level (Li et al, 2021; Zhang et al, 2022). Severe fever with thrombocytopenia syndrome is an emerging infectious disease caused by Severe Fever with Thrombocytopenia Syndrome Virus (SFTSV) transmitted by tick bites. SFTSV, officially named *Dabie bandavirus*, belongs to the family *Phenuiviridae* of the order *Bunyavirales* and is a negative-stranded RNA virus with segmented genomes. SFTSV infection can cause severe clinical symptoms, including hemorrhagic fever, thrombocytopenia, leukocytopenia, gastrointestinal symptoms, and a high case fatality rate of 12–30% (Zhou and Yu, 2021). The primary vector of SFTSV is *Haemaphydalis longicornis* tick, widespread in East Asia, Oceania, and North America. Since its isolation in China in 2009 (Yu et al, 2011), infection cases have been documented in South Korea, Japan, and Vietnam and shown expanding geographic distribution (Li et al, 2021). SFTSV genomes contain three segments designated as large (L), medium (M), and small (S) segments, respectively. The L segment encodes an RNA-dependent RNA polymerase (RdRp) that mediates transcription and replication of the viral genome. The M segment encodes a glycoprotein precursor (Gp) that is cleaved by cellular proteases during translation and processed into two subunits: glycoprotein N (Gn) and glycoprotein C (Gc), which mediate entry and assembly of the virion (Hofmann et al, 2013). The S segment encodes a nucleoprotein (NP) that mediates the formation of the viral ribonucleoprotein (RNP) complex and a nonstructural protein (NSs), which serves as an essential virulent factor (Sun et al, 2018).

Nucleotide modification plays critical regulatory roles in transcription and translation in eukaryotes responding to environmental impact. N6-methyladenosine (m6A) is the most abundant internal modification of eukaryotic RNAs, which can influence RNA structure, localization, and functions. Therefore, m6A modification regulates many biological processes, such as development (Fustin et al, 2013; Zhong et al, 2018), metabolism (Lence et al, 2016; Xiong et al, 2021) and tumorigenesis (Yu et al, 2021b; Jiang et al, 2021). Numerous studies reported that viral transcripts or genomic RNAs contained m6A sites and the viral replication or the functions were influenced by m6A modification (Lichinchi et al, 2016b, 2016a; Brocard et al, 2017). In a recent study, Wang et al analyzed samples from seven SFTSV-positive patients and unraveled the potential relationship between m6A epigenetic modification of SFTSV with a high fatality rate (Wang et al, 2021). However, the specific role of m6A in SFTSV RNA has not been addressed.

[1]Center for Public Health Research, Medical School of Nanjing University, Nanjing, People's Republic of China. [2]Department of Infectious Diseases, Nanjing Drum Tower Hospital, Affiliated Hospital of Medical School, Nanjing University, Nanjing, People's Republic of China. [3]Clinical Molecular Diagnostic Laboratory, The 2nd Affiliated Hospital of Nanjing Medical University, Nanjing, China. [4]State Key Laboratory of Analytical Chemistry for Life Science, Nanjing University, Nanjing, People's Republic of China. [5]Yunnan Provincial Key Laboratory of Entomological Biopharmaceutical R&D, College of Pharmacy, Dali University, Dali, People's Republic of China. ✉E-mail: wzhw@nju.edu.cn

m6A modification is installed and regulated by "writers", namely, the methyltransferase (MTase) complex methyltransferase-like 3 (METTL3)/ METTL14/Wilms'-tumor-1-associating protein (WTAP) (Bokar et al, 1997; Liu et al, 2014; Ping et al, 2014; Zhou and Pan, 2016), and "eraser" proteins including demethylase fat mass and obesity-associated gene (FTO) (Jia et al, 2011) and AlkB homolog 5 (ALKBH5) (Zheng et al, 2013). The fate of m6A-methylated RNAs is mainly mediated through "reader" proteins YT521-B homology domain family, including YTHDF1, YTHDF2, and YTHDF3 et al, which can read the m6A modification sites on RNAs and mediate a series of downstream reactions. Several studies found that YTHDF1 enhanced the translational efficiency of m6A-modified mRNAs (Wang et al, 2015). In contrast, Zaccara et al found no evidence for the role of YTHDFs in promoting translation; instead, they found that YTHDFs acted redundantly to induce the degradation of the same subset of mRNAs (Zaccara and Jaffrey, 2020). Xia et al demonstrated that YTHDF1 recognized and destabilized m6A-modified EBV transcripts (Xia et al, 2021). Thus, the functions of YT521-B family are complex and diverse, which are distinct in different cells and cannot be thoroughly defined.

Protein post-translational modifications (PTMs), such as protein methylation, phosphorylation, and ubiquitination, play important roles in immune response, signaling pathway regulation, and viral immune escape. Lactylation is a novel lactate-derived PTM on the lysine residue of protein (Zhang et al, 2019) and functions as a vital epigenetic regulator in many cellular processes. Xiong et al identified the lactylation of lysine residues 281 and 345 of METTL3, which strengthened the immunosuppressive functions of myeloid cells (Xiong et al, 2022). Yu et al revealed that histone lactylation facilitated YTHDF2 expression. Moreover, YTHDF2 recognizes the m6A-modified PER1 and TP53 mRNAs and promotes their degradation, which accelerates tumorigenesis of ocular melanoma (Yu et al, 2021b). However, while there have been reports of SUMOylation (Sugiokto et al, 2024), O-GlcNAcylation (Li et al, 2022a), ubiquitination (Wang et al, 2023a), and other modifications on YTHDF1, no other novel lysine post-translational modifications of YTHDF1 were reported and how a virus uses post-translational protein modifications to degrade m6A modification-related proteins, thereby promoting its own replication.

In this study, we reported the presence of m6A modification in SFTSV RNAs and examined the influence and regulation of host m6A modification on SFTSV infection and replication. We have, for the first time, identified lactylation modification sites on YTHDF1 and demonstrated that SFTSV exploited this feature to facilitate its replication.

# Results

## All SFTSV genomic segments contain m6A modifications

m6A modifications are widely present in viruses. Wang et al reported m6A modification in the M segment of SFTSV originating from Henan (Wang); however, the roles of the modifications were not studied, and the S and L RNA segments have not been reported to contain m6A modifications. Furthermore, there have been no reports on whether there are differences in modifications among other prevalent strains of SFTSV in China. We analyzed m6A modification patterns of SFTSV S, M, and L segments of a subtype E SFTSV strain, E-JS2013-24, a strain isolated from an acutely infected individual in Jiangsu Province, which were extensively investigated in our previously published works (Ji et al,

2024; Song et al, 2018; Wu et al, 2020). HeLa cells were infected with JS2013-24 SFTSV for 36 h, and whole cellular RNA was prepared for MeRIP-seq analysis (Fig. 1A). m6A-modified peaks were identified in the L, M, and S RNAs of SFTSV (Fig. 1B). Sequencing results indicated that the S segment has the highest level of m6A modification, followed by the M segment, and the L segment has the lowest level of m6A modification (Fig. 1B). Meanwhile, we found that the abundance and location of m6A modifications slightly differed between Henan and Jiangsu strains. These differences may be the result of differences between in vivo and in vitro infections or viral evolution. Nevertheless, this indicates that m6A modification is a dynamic and variable epigenetic modification during viral infection, which may affect viral evolution or virulence.

As an RNA virus, SFTSV replicates in the cytoplasm, we first examined whether the m6A machinery could successfully modify SFTSV RNAs. Immunofluorescence results revealed that both METTL3 and ALKBH5 aggregated in the cytoplasm following infection (Fig. EV1A,C), while METTL14 remained localized in the nucleus before and after infection (Fig. EV1B). It is evident that METTL14 is unlikely to play a role in modifying SFTSV RNAs. Next, through nuclear-cytoplasmic fractionation experiments, we found that both METTL3 and ALKBH5 were present in the cytoplasm before and after SFTSV infection (Fig. 2A). In addition, there was a slight decrease in nuclear METTL3 levels after infection, accompanied by a corresponding increase in cytoplasmic METTL3. This phenomenon suggests that METTL3 and ALKBH5 are highly likely to modify SFTSV RNAs. We validated this hypothesis through METTL3-IP-qRT-PCR and ALKBH5-IP-qRT-PCR experiments (Fig. 2B–G). Next, we used SRAMP to predict potential m6A modification sites within the three RNA segments and intersected these predictions with the MeRIP-seq results, identifying eight high-confidence sites (Table EV4). We then performed A to U mutant on these eight sites to identify real m6A modification sites. The results from m6A-IP-qRT-PCR indicated that position A2531 in the L segment was a false-positive m6A modification site (Fig. 2J). In contrast, mutations at the other seven sites led to a significant reduction in the amount of pulled-down RNA, confirming their roles as m6A modification sites (Fig. 2H,I,K–O).

## SFTSV infection downregulated the level of m6A reading protein YTHDF1

Because there has been no previous research on the roles of m6A modification on SFTSV infection, we firstly investigated the impact of methyltransferases and demethylases on SFTSV. Knockdown of METTL3 with siMETTL3 did not affect viral replication as shown by NP expression (Fig. EV1D) while knockdown of ALKBH5 caused significant reduction of SFTSV NP expression (Fig. EV1E). While the overexpression of METTL3 significantly downregulated the replication of SFTSV (Fig. EV1F), and overexpressed ALKBH5 did not show significant changes (Fig. EV1G), suggesting that m6A modification adversely affected the viral replication. Intriguingly, we found that YTHDF1 protein but not YTHDF2/3 (Fig. EV2A) or other m6A machinery (Fig. EV2B–E) decreased in a time- and viral dose-dependent manner in infected HeLa and HUVEC cells (Fig. 3A–D) but showed no significant changes at the mRNA level (Fig. 3E–H). We speculated that, as a reader protein, YTHDF1 would identify m6A modifications upon on SFTSV RNAs, facilitate downstream process of SFTSV mRNAs, thus affecting viral replication. The downregulation of YTHDF1 responding to time of infection and viral inoculum indicated a virus-driven process.

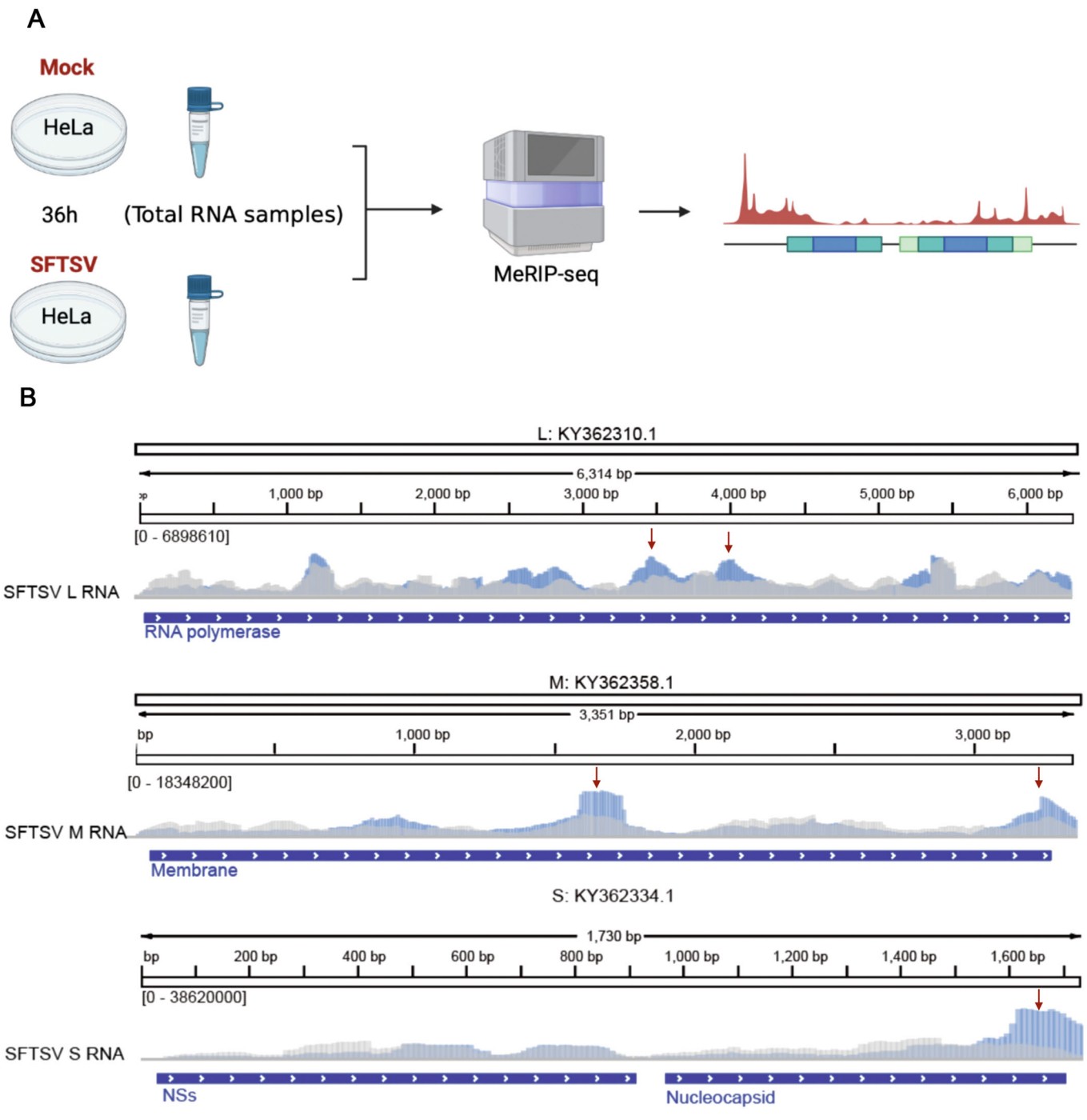

**Figure 1. The SFTSV RNAs are m6A-modified.**

(A) The flow chart shows the main flow of MeRIP-seq sequencing. HeLa cells uninfected or infected with SFTSV (MOI = 1) were harvested for total RNA at 36 h (Created with BioRender.com). (B) Distribution of m6A peaks in SFTSV RNAs. A schematic diagram of the SFTSV L, M, and S RNA was shown. m6A-seq of SFTSV RNA, showing the distribution of m6A immunoprecipitation reads (blue block) mapped to SFTSV RNAs. The baseline signals from input samples are shown as a gray line. The positive signals were pointed with red arrows. Source data are available online for this figure.

Consistently, knockdown YTHDF1 upregulated the expression of both NP and NSs (Fig. 3I), and the expression of exogenous YTHDF1 downregulated the expression of NP and NSs (Fig. 3J). These results demonstrated that YTHDF1 negatively regulated the translational efficiency of SFTSV mRNAs.

## SFTSV infection increased lactylation of YTHDF1

To explore the mechanism of YTHDF1 degradation after SFTSV infection, mock-infected or SFTSV-infected HeLa cells were harvested after 36 h and subjected to YTHDF1-Co-IP. Western blot analysis was

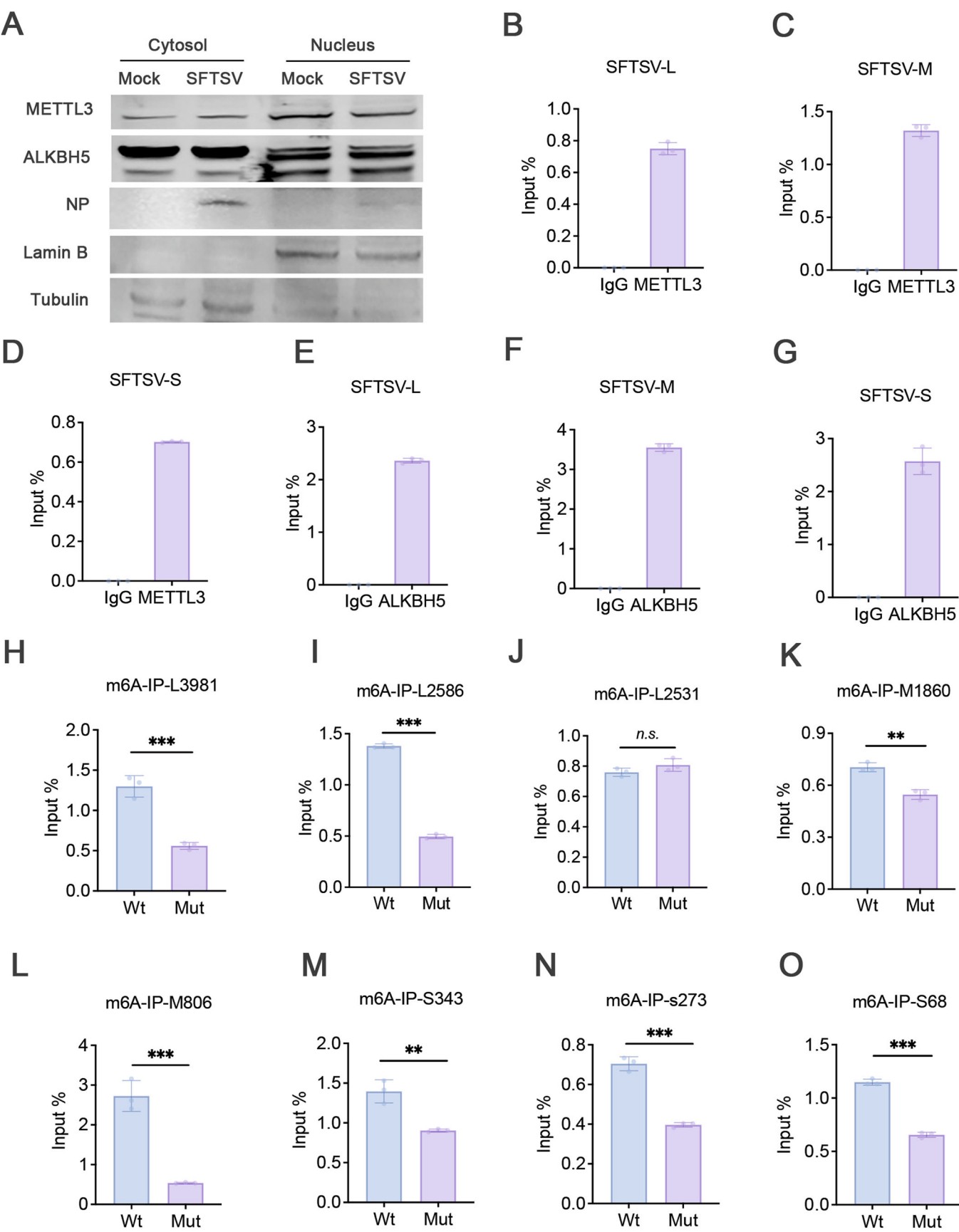

Figure 2.  Identification of m6A modification sites.

(A) The distribution of METTL3 and ALKBH5 were assessed through nuclear-cytosol fractionation experiment. The expression levels of METTL3 and ALKBH5 were analyzed by western blot. Lamin B was used as the loading control for the nuclear fraction, while Tubulin served as the loading control for the cytoplasmic fraction. There're isoforms of ALKBH5 with MW 40-44 kDa and 52 kDa. (B–D) HeLa cells were infected with SFTSV for 24 h. METTL3-IP-qRT-PCR were performed to analyze SFTSV RNAs. The fold enrichment was determined by calculating the 2-Δct of the sample relative to input. (E–G) HeLa cells were infected with SFTSV for 24 h. ALKBH5-IP-qRT-PCR were performed to analyze SFTSV RNAs. The fold enrichment was determined by calculating the 2-Δct of the sample relative to input. (H–O) HEK-293T cells were transfected with Wild-type plasmids or site-mutant plasmids for 24 h. m6A-IP-qRT-PCR were performed to analyze m6A-modified RNAs. The fold enrichment was determined by calculating the 2-Δct of the sample relative to input. The results are represented as the mean ± SD of $n = 3$ biological replicates. Statistical significance was determined by a two-sided Student's $t$ test (L3981: ***$P = 0.0008$, L2586: ***$P < 0.0001$, L2531: n.s. $= 0.1673$; M806: ***$P = 0.0006$, M1860: **$P = 0.0020$; S68: ***$P < 0.0001$, S273: ***$P = 0.0001$, S343: **$P = 0.0044$). Source data are available online for this figure.

performed using six pan-antibodies specific for various post-translational modifications, including lysine 2-hydroxyisobutyrylation (Khib), lysine crotonylation (Kcr), lysine borylation (Kbu), lysine lactylation (Klac), lysine propionylation (Kpr), and lysine acetylation (Kac), respectively. The results showed that compared with the non-specific IgG group, the Klac modification of YTHDF1 was detected in both mock and infected cells, and the level of Klac modification increased after SFTSV infection (Fig. 4A). Similarly, we observed the same phenomenon in HUVEC cells (Fig. 4B), suggesting that it is not cell specific. In addition, we found that total lactate levels were significantly upregulated after SFTSV infection (Fig. 4C). It is known that sodium L-lactate can significantly elevate intracellular lactate levels and increase the lactylation of lactylation-modified proteins (Xiong et al, 2022). We found that the lactylation of YTHDF1 was increased after the cells being treated with Sodium L-lactate (Fig. EV3C). We next investigated whether Klac modification would affect YTHDF1 stability. HeLa and HUVEC cells were treated with 10 mM sodium L-lactate to increase the lactylation level of cellular proteins, followed by treatment with protein synthesis inhibitor CHX to stop the synthesis of new YTHDF1 and YTHDF1 stability was examined. The results indicated that the addition of sodium L-lactate followed with CHX treatment resulted in more than 50% reduction of YTHDF1 in 4 h (Fig. 4D,E), suggesting that the overall lactylation significantly affected YTHDF1 protein stability. To determine lactylation sites on YTHDF1, we predicted specific Klac sites with DeepKla (lin-group.cn/server/DeepKla/index) and identified five high-confidence potential lactylation sites on YTHDF1 protein sequence (Table EV5). To identify the critical lactylation site of YTHDF1, lactylation-defective mutants (K to R) of these five lactylation sites were generated. Through flag-IP we found that YTHDF1-K517R and YTHDF1-K521R mutants showed reduced lactylation (Fig. 4F), indicating that K517 and K521 might be essential lactylation sites of YTHDF1. In a previous study, Wang et al found that YTHDF1 could be modified by ubiquitination (Wang et al, 2023a). We thus co-expressed HA-ub and K517R-YTHDF1 or K521R-YTHDF1 to investigate whether mutation of the lactylation site would affect the ubiquitin level on YTHDF1. The results showed that the ubiquitin levels of K517R-YTHDF1 and K521R-YTHDF1 proteins were significantly lower than that of wild-type YTHDF1 (Fig. 4G). This result indicated that lactylation modification can lead to an increased ubiquitin level of YTHDF1, thereby affecting the protein stability of YTHDF1.

## Sirt6 and ESCO1 are eraser and writer, respectively, for Klac modification of YTHDF1

The histone deacetylases HDAC type I (1-3) and the family of Sirtuins type III, have been identified to mediate protein delactylation (Moreno-Yruela et al, 2022). Next, we determined which acetyltransferases mediated the lactylation of YTHDF1 by overexpressing deacyltransferases Sirtuins family members, Sirt1, Sirt2, Sirt3, Sirt4, Sirt5, Sirt6 and Sirt7, respectively, in HEK-293T cells and determining Klac modification levels by YTHDF1-Co-IP. We observed that all members of the Sirtuins family, except for Sirt5, were co-precipitated with YTHDF1, and overexpression of only Sirt4 and Sirt6 resulted in a downregulation of Klac modification on endogenous YTHDF1 (Fig. 5A), suggesting that though all Sirtuins except Sirt5 were associated with YTHDF1 only Sirt4 and Sirt6 participated in delactylation of YTHDF1.

We observed that the SFTSV nonstructural protein NSs, but not NP, co-localized with YTHDF1 (Fig. EV3A,B). Previous studies reported that SFTSV NSs could form inclusion bodies in the cytoplasm and hijack host molecules into the inclusion bodies to inhibit antiviral signaling (Wu et al, 2014). Therefore, we investigated the roles of NSs in the lactylation of YTHDF1 by co-transfecting NSs with either Sirt4 (Fig. 5B) or Sirt6 (Fig. 5C) in HEK-293T cells and performing co-IP assay to determine protein interactions. We detected co-immunoprecipitation between NSs and Sirt6, but not Sirt4. Next, we investigated how the interaction of NSs and Sirt6 regulated the function of Sirt6. By immunofluorescence, we observed that overexpression of NSs induced nuclear translocation of Sirt6, resulting in the colocalization of Sirt6 and NSs in the cytoplasm (Fig. 5D). However, we did not find that SFTSV infection downregulated Sirt6 (Fig. 5F). Co-IP of YTHDF1 and GFP-NSs in HEK-293T cells showed that the overexpression of NSs downregulated the formation of YTHDF1 and Sirt6 complex (Fig. 5E), and increased the Klac level of YTHDF1 (Fig. 5E). We also found that the stability of YTHDF1 protein was increased by ectopically expressed Sirt6 in cells treated with sodium L-lactate (Fig. 5G). The above findings suggest that Sirt6 is a delactyltransferase catalyzing the removal of Klac from YTHDF1.

We next determined which acetyltransferases of EP300, ESCO1, ESCO2, MYST1, or ATAT1 catalyzed YTHDF1 lactylation, and whether NSs protein directly impacted on the acetyltransferases. We performed YTHDF1-Co-IP of exogenously expressed EP300, ESCO1, ESCO2, MYST1, or ATAT1 in HEK-293T cells and measured the changes in Klac modification levels (Fig. EV4A) and observed that overexpression of EP300, ESCO1 or ATAT1 upregulated the levels of Klac modification of YTHDF1. However, only ESCO1 was associated with YTHDF1 and with SFTSV NSs (Fig. EV4B,C), suggesting that EP300- and ATAT1-mediated lactylation may be through indirect or a non-NSs-mediated mechanism. We than found that SFTSV infection had no effect on ESCO1 expression (Fig. EV4D), and the interaction between NSs and ESCO1 didn't affect the formation of YTHDF1 and ESCO1 complex (Fig. EV4C).

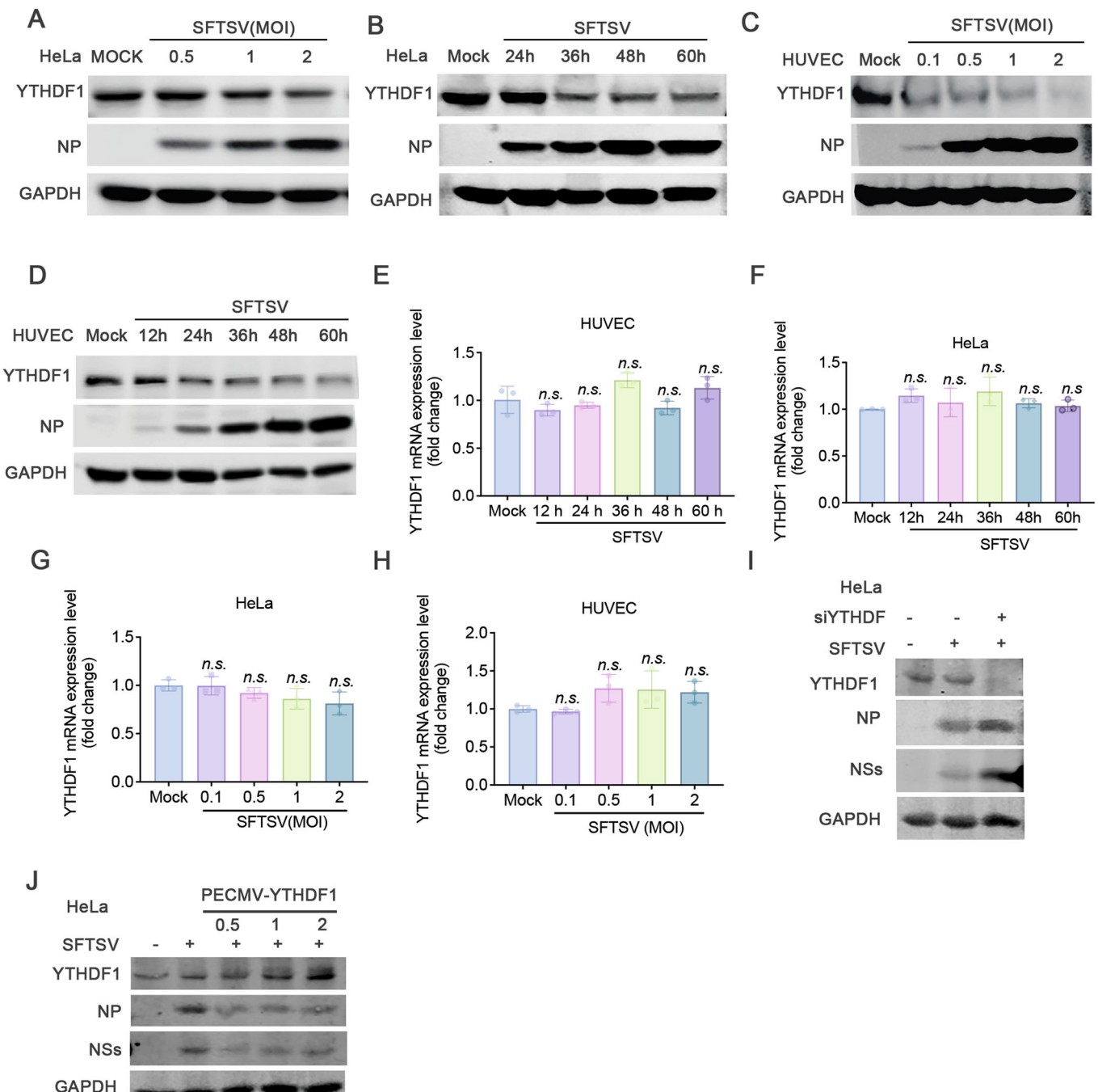

**Figure 3. SFTSV infection downregulated the expression of m6A reading protein YTHDF1.**

(A) HeLa cells and HUVEC cells (C) were infected with SFTSV in a dose-dependent manner, and then the expression level of YTHDF1 were detected by western blot. GAPDH was used as a loading control. (B) HeLa cells and HUVEC cells (D) were infected with SFTSV (MOI = 1) in a time-dependent manner, and then the expression level of YTHDF1 were detected by western blot. GAPDH was used as a loading control. (E) HeLa cells and HUVEC cells (F) were infected with SFTSV (MOI = 1) in a time-dependent manner, and then the expression level of YTHDF1 RNA were detected by qPCR. The fold enrichment was determined by calculating the 2-Δct of the sample relative to the GAPDH. The results are represented as the means ± SD of *n* = 3 biological replicates. Statistical significance was determined by a two-sided Student's *t* test (n.s. = 0.2907, 0.5317, 0.0936, 0.4005, 0.3082). (G) HeLa cells and HUVEC cells (H) were infected with SFTSV in a dose-dependent manner, and then the expression level of YTHDF1 RNA were detected by qPCR. The fold enrichment was determined by calculating the 2-Δct of the sample relative to the GAPDH. The results are represented as the means ± SD of *n* = 3 biological replicates. Statistical significance was determined by a two-sided Student's *t* test (n.s. = 0.46, 0.0973, 0.0871, 0.3674). (I) The HeLa cells were transfected with siYTHDF1 or siNC (Negative control) for 36 h and followed infect with SFTSV 36 hpi. The expression levels were detected by western blot. GAPDH was used as a loading control. (J) The HeLa cells were transfected with 0.5 μg, 1 μg, and 1.5 μg YTHDF1 or control plasmid for 36 h and followed infected with SFTSV 36 hpi. The expression levels were detected by western blot. GAPDH was used as a loading control. Source data are available online for this figure.

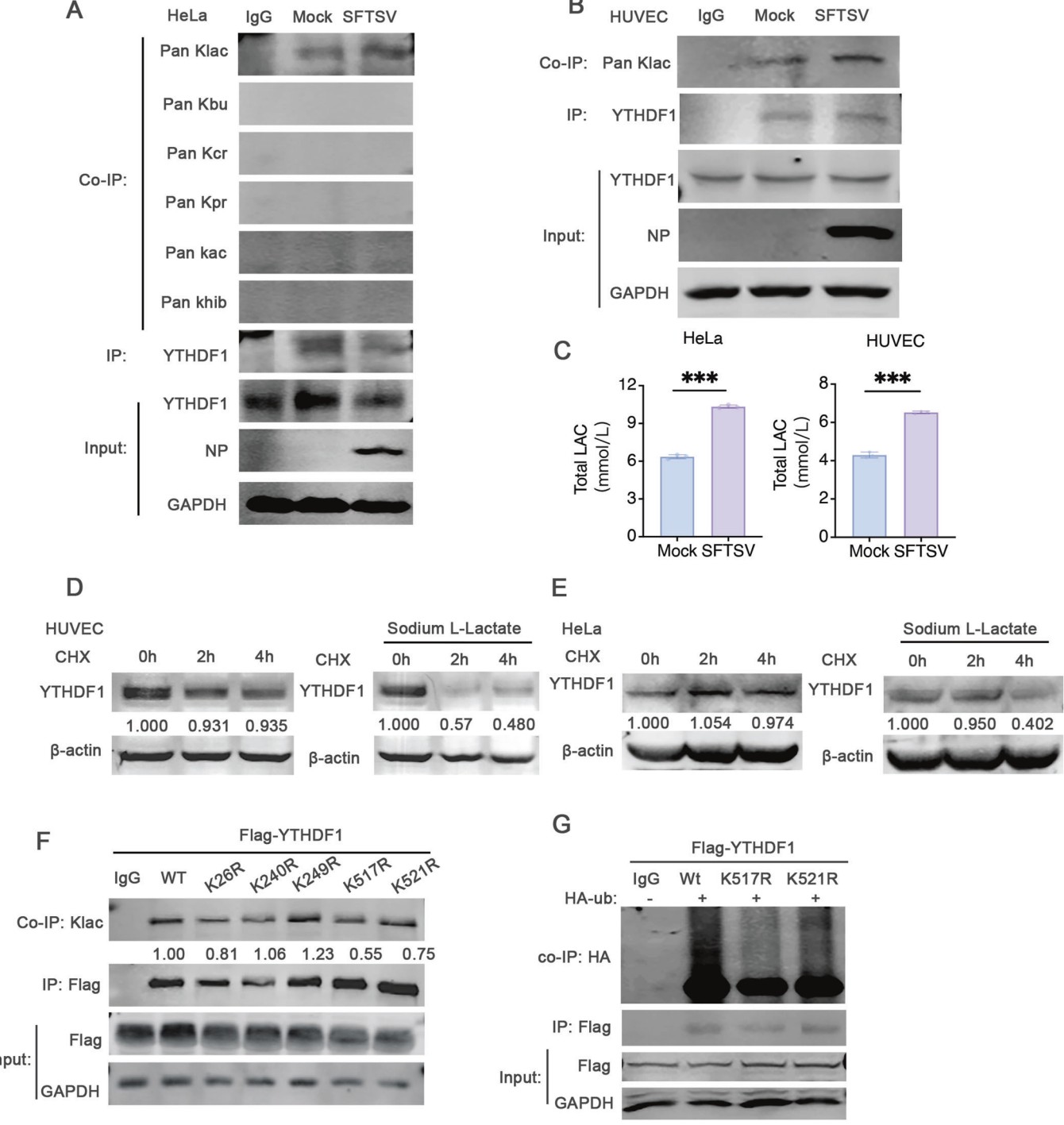

## m6A modification mediated the interaction between YTHDF1 and SFTSV RNAs

The above observation suggested a connection between the decrease in YTHDF1 and the increased intracellular lactate level, along with a corresponding rise in YTHDF1 lactylation. As a m6A reading protein, YTHDF1 was reported to recognize m6A sites to mediate various biological functions (Jiang et al, 2021). Thus, to illustrate the role of YTHDF1 during SFTSV infection, whole cell RNAs of SFTSV-infected

HeLa were isolated and subjected to RNA immunoprecipitation (RIP) by an YTHDF1-specific antibody followed by qPCR. Notably, all three segments of the SFTSV RNAs were associated with YTHDF1 and the association was dependent on the abundance of m6A modifications on SFTSV RNAs since knockdown of METTL3 with specific siRNAs would significantly disrupt the associations (Fig. 6A). To further investigate the regulatory mechanism of YTHDF1 in SFTSV infection, HeLa cells were transfected with YTHDF1 siRNA or siNC (Negative control), and subsequently infected with SFTSV. The results showed that knockdown

◄

**Figure 4. SFTSV infection increased lactylation levels of YTHDF1.**

(A) Endogenous YTHDF1 in uninfected or SFTSV-infected HeLa cells was collected by co-IP assay using YTHDF1-specific antibody, and then detected lysine modifications on YTHDF1 protein with pan-antibodies (Klac, Kbu, Kcr, Kpr, Kac, and Khib). Non-specific IgG antibody was used as negative control. (B) Klac modification was detected on YTHDF1 by performing endogenous YTHDF1- IP in uninfected or SFTSV-infected HUVEC cells. Non-specific IgG antibody was used as negative control. (C) Cell supernatants and whole cell lysates were collected and tested total lactate concentration in uninfected or SFTSV-infected (MOI = 1) HeLa and HUVEC cells. The results are represented as the means ± SD of $n = 3$ biological replicates. Statistical significance was determined by a two-sided Student's $t$ test (HeLa: Mock vs. SFTSV $P < 0.0001$; HUVEC: Mock vs. SFTSV $P < 0.0001$). (D) HeLa and (E) HUVEC cells were treated with or without 10 mM Sodium L-Lactate. The cells were treated with the protein synthesis inhibitor cycloheximide (CHX; 100 μg/mL) for the indicated periods before harvesting. β-actin was used as a loading control. (F) HEK-293T cells transfected with indicated wild-type-YTHDF1 or sites-mutant YTHDF1 and were collected by co-IP using Flag specific antibody, and the lactylation level of all YTHDF1 were detected by anti-pan-lactyl antibody. (G) HEK-293T cells co-transfected with K517R-YTHDF1, K521R-YTHDF1 or Wt-YTHDF1 with HA-ub and collected by co-IP using Flag specific antibody, and ubiquitination of all YTHDF1 were detected by anti-HA antibody. Source data are available online for this figure.

YTHDF1 by siYTHDF1 had no effect on the L, M, and S RNAs of SFTSV (Fig. EV5A–C), while overexpression of exogenous YTHDF1 significantly inhibited the expression of all three SFTSV RNA segments, with a dose-dependent effect (Fig. 6B). Next, we determined if YTHDF1 affected the stability of SFTSV RNAs. Before investigating the effect of YTHDF1 on SFTSV RNA stability, we first confirmed whether the interaction between YTHDF1 and SFTSV RNA is mediated by m6A modifications. In this study, we focused on the S segment, and therefore selected the three m6A modification sites within the S RNA for YTHDF1-qRT-PCR analysis, with L2531 serving as a negative control (Fig. EV5D). The results indicated that A63 and A343 are binding sites for YTHDF1 (Fig. EV5E,F), while A273, although an m6A modification site, may not be a primary binding site for YTHDF1 (Fig. EV5G). Then, we co-transfected HEK-293T cells with PECMV-YTHDF1 together with NP or Gn plasmid, followed with 5 μg/mL actinomycin D (RNA Polymerase II inhibitor) treatment and measured the half-life of SFTSV NP mRNA by qPCR. The results indicated that YTHDF1 binding significantly reduced the stability of SFTSV NP mRNA (Fig. 6C), which was corroborated by the increased NP and NSs expression in SFTSV-infected cells with YTHDF1 knock-down (Fig. 3I,J). To validate our hypothesis, we synthesized wild-type and m6A-site mutant (A68T, A273T, and A343T) SFTSV NP dual-luciferase reporter plasmids and co-transfected the cells with these constructs along with the YTHDF1 plasmid. Bioluminescent analysis of the HEK-293T cells showed that the exogenous YTHDF1 markedly inhibited the luciferase activity of SFTSV NP, while the inhibition was notably abrogated when the m6A sites were mutated (Fig. 6D). These data together demonstrated that YTHDF1 decreased the RNA stability and protein translational efficiency of SFTSV S segment in an m6A-dependent manner.

## Discussion

m6A modification is one of the most widely occurred epigenetic modifications in eukaryotes. Upon virus entry into the host, viral RNA is also modified and regulated by the m6A mechanisms of the host cells, as part of the host defense against viral infection (Dang et al, 2019; Ma et al, 2021); As a counteract, viruses evolve various mechanisms to evade or utilize the epigenetic modifications to their advantages (Lu et al, 2020). In this study, we found that all three segments of the SFTSV genomic RNAs and mRNAs contain m6A modifications (Figs. 1B and 2).

The inhibition of SFTSV RNA replication and viral protein synthesis by overexpression of YTHDF1 (Fig. 6B) suggests that YTHDF1 acts as a negative regulator of viral RNAs. YTHDF1, one of the m6A-binding proteins, is recruited to m6A sites and impacts on the fate of m6A-modified RNA. Knowledge on the roles of

YTHDFs is increasing rapidly and three hypothetical models have been proposed: the Canonical model suggests that specific proteins regulate specific m6A-RNAs (Wang et al, 2015, 2014); the Competing model proposes that all YTHDFs mediate the degradation of m6A-RNAs (Zaccara and Jaffrey, 2020); and the Combined model posits that YTHDFs interact with various forms of partner proteins, determining the different fates of m6A-RNAs (Zou et al, 2023). Although we found that both knocking down and over-expressing YTHDF1 significantly affected the translational efficiency and protein expression of SFTSV (Fig. 3I,J), only overexpression of YTHDF1 led to a dose-dependent decrease in SFTSV RNAs (Fig. 6B). We speculated that this could be due to excessive YTHDF1 binding to SFTSV RNAs, leading to the formation of LLPS (Liquid-Liquid Phase Separation) (Zhu et al, 2014; Li et al, 2022b), resulting in decreased RNA stability. Additionally, Due to SFTSV infection-induced downregulation of YTHDF1, siYTHDF1 may have an insignificant effect on SFTSV RNA.

SFTSV infection downregulated the protein level of reader protein YTHDF1 without impacting on YTHDF1 RNAs (Fig. 3E–H), suggesting a post-translational mechanism. Previous studies reported that m6A demethylase ALKBH5 and methylase METTL3 contained novel protein PTM sites (Yu et al, 2021a; Du et al, 2018) and several types of PTMs, e.g., glycosylation (Li et al, 2022a) and ubiquitination (Wang et al, 2023a), have been reported to regulate YTHDF1 activity. In the current study, we identified a novel lysine modification, Klac of YTHDF1 and showed that the lactylation of YTHDF1 was enhanced by SFTSV infection (Fig. 4A). The lactylation of YTHDF1 reduced the stability of YTHDF1 (Fig. 4D). Point mutation showed that either K517 or K521 had the most pronounced effect on Klac of YTHDF1 while other potential sites had no effect (Fig. 4F), indicating that these two lysines were the sites of lactylation.

Zhen et al reported recently that HDAC1–3 and Sirt1-3 could catalyze Klac modification of histones (Moreno-Yruela et al, 2022); A recent study proposed that histone lactylation can directly promote gene transcription and induced the polarization of macrophages from M1 to M2 (Zhang et al, 2019). However, the exploration of the role of lactylation on nonhistone is also crucial, and the enzymes responsible for Klac depositon or removal in nonhistone proteins also need to explore. Wang et al found that the lactylation at K183 of YY1 enhanced retinal neovascularization, and identified p300 as a lactylation "eraser" on YY1 (Wang et al, 2023b). Pang et, al. identified the lactylation elevation of HSPA6 during PRRSV infection, which affected the IFN-β production (Pang et al, 2024). We

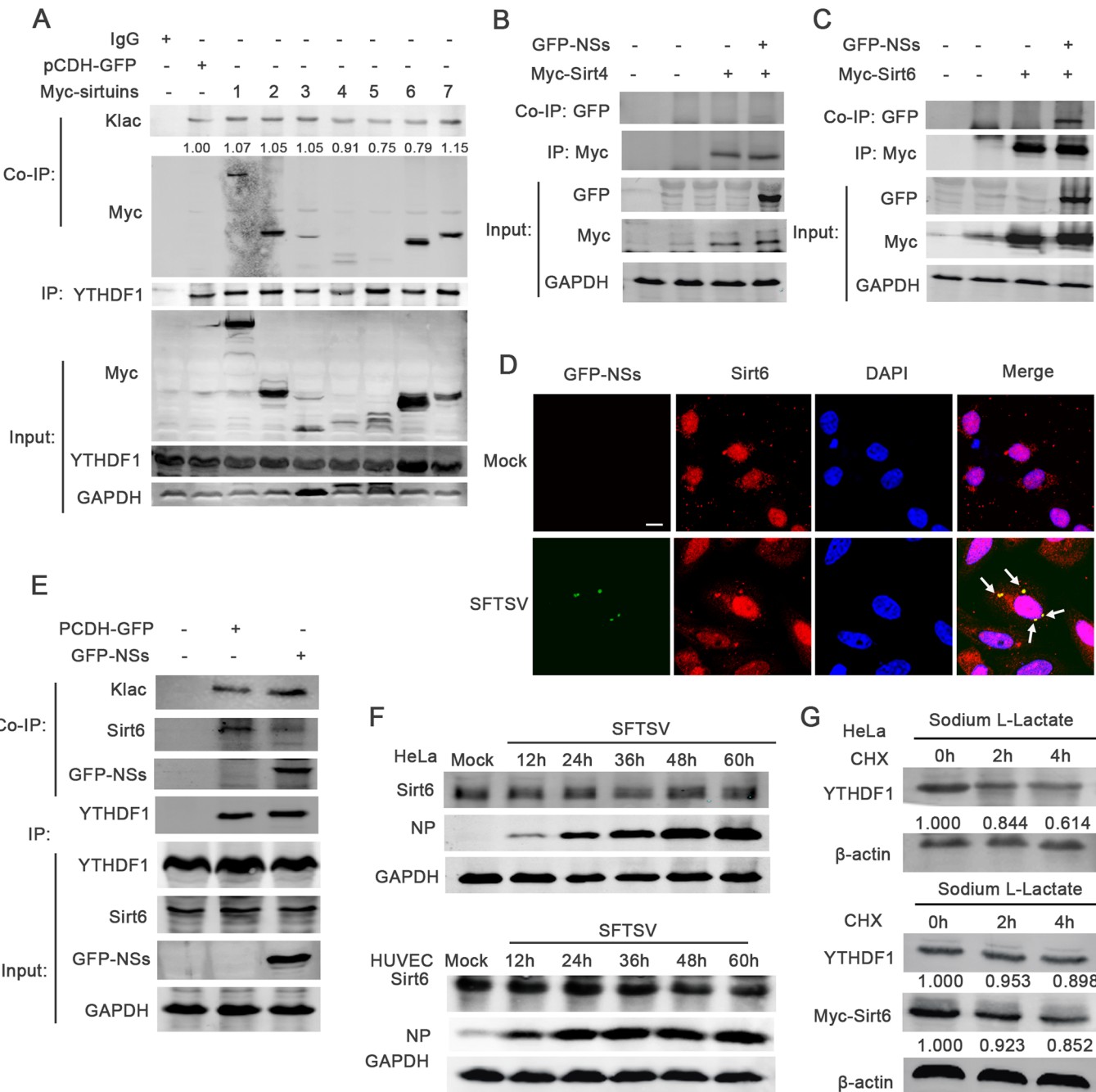

**Figure 5. Overexpression of SFTSV NSs decreased the binding affinity of YTHDF1 and Sirt6.**

(A) The endogenous YTHDF1 was collected by co-IP using YTHDF1-specific antibody in HEK-293T cells. The interaction between YTHDF1 and Myc-Sirtuins 1-7 were detected by using Myc-specific antibody, and the lactylation level were detected by anti-Klac antibody. (B) The interaction between GFP-NSs and Myc-Sirt4 was detected by IP using Myc-specific antibody in HEK-293T cells. (C) The interaction between GFP-NSs and Myc-Sirt6 was detected by IP using Myc-Specific antibody in HEK-293T cells. (D) Immunofluorescence assay were performed to measure the distribution of Sirt6 (Red) and NSs (green) in HeLa cells after SFTSV infection. The white arrows indicate the colocalization. Cell nuclei were stained with DAPI. Scale bar = 10 μm. (E) HEK-293T cells was transfected with or without GFP-NSs plasmid, and endogenous co-IP was performed to measure the lactylation levels of YTHDF1 and the interaction between endogenous YTHDF1 and Sirt6 by using anti-klac antibody, anti-YTHDF1 antibody, and anti-Sirt6 antibody. (F) The expression of Sirt6 protein was detected in SFTSV (MOI = 1) infected HeLa and HUVEC cells in a time-dependent manner by western blot. GAPDH was used as a loading control. (G) HeLa cells were transfected with or without Myc-Sirt6 plasmid and then treated with Sodium L-Lactate. The cells were treated with the protein synthesis inhibitor cycloheximide (CHX; 100 μg/mL) for the indicated periods before harvesting. β-actin was used as a loading control. Source data are available online for this figure.

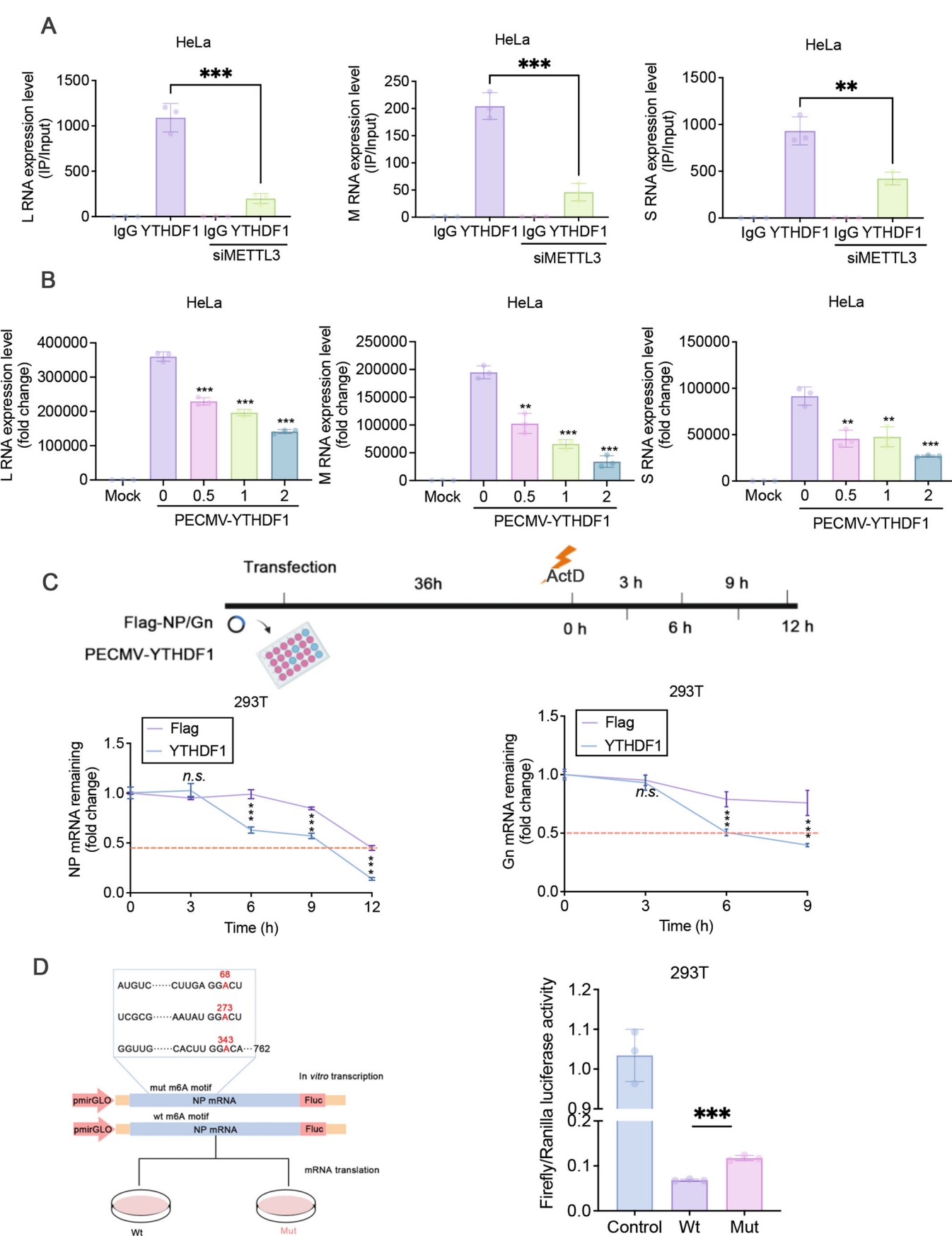

**Figure 6. m6A modification mediated the interaction between YTHDF1 and SFTSV RNAs.**

(A) HeLa cells were transfected with siNC (Negative control) or siMETTL3 for 24 h, and then infected with SFTSV (MOI = 1). The L, M, and S RNA of SFTSV were collected by RIP at 36 h using YTHDF1-specific antibody, and the relative expression level were detected by qPCR. The fold enrichments were determined by calculating the 2-Δct of the RIP sample relative to the input sample. The results are represented as the means ± SD of $n = 3$ biological replicates. Statistical significance was determined by a two-sided Student's $t$ test (***$P = 0.0007$, ***$P = 0.0007$, **$P = 0.0058$). (B) HeLa cells were transfected with or without 0.5 µg, 1 µg, or 2 µg plasmid of YTHDF1, and then infected with SFTSV (MOI = 1). The expression level of L, M, and S RNA of SFTSV was detected at 36 h by qPCR. The fold enrichment was determined by calculating the 2-Δct of the sample relative to the GAPDH. The results are represented as the means ± SD of $n = 3$ biological replicates. Statistical significance was determined by a two-sided Student's $t$ test (L: ***$P = 0.0002$, ***$P < 0.0001$, ***$P < 0.0001$; M: ***$P = 0.0018$, ***$P < 0.0001$, ***$P < 0.0001$; S: **$P = 0.0041$, **$P = 0.0065$, **$P = 0.0003$). (C) HEK-293T cells were co-transfected plasmids PECMV-YTHDF1 with NP or Gn, followed the treatment of ActD. The residual mRNA levels were collected every 3 h, and detected by qPCR. The relative mRNA level at 0 h after the ActD treatment was normalized to 1. The results are represented as the means ± SD of $n = 3$ biological replicates. Statistical significance was determined by two-way ANOVA (NP: n.s. > 0.9999, n.s. = 0.1024, ***$P < 0.0001$, ***$P < 0.0001$; Gn: n.s. > 0.9999, n.s. = 0.9843, ***$P < 0.0001$, ***$P < 0.0001$). (D) YTHDF1-overexpression HEK-293T cells were transfected with the m6A-mutated NP plasmid or WT-NP plasmid for 36 h; the translation effect was determined by a dual-luciferase reporter system. The mutated sites have been indicated in the schematic diagram. The results are represented as the means ± SD of $n = 3$ biological replicates. Statistical significance was determined by a two-sided Student's $t$ test (***$P = 0.0002$). Source data are available online for this figure.

further identified ESCO1 and Sirt6 as a writer and eraser of YTHDF1 lactylation, respectively, and showed that mutating the lactylation sites reduced the ubiquitination of the YTHDF1 (Fig. 4G), consistent with a previous report that there is a crosstalk between lactylation and ubiquitination (Wu and Tao, 2022), providing a mechanistic explanation for the virus-induced degradation of YTHDF1.

NSs, a SFTSV virulence factor, has been shown to play diverse roles in modulating host response to infection and pathogenic manifestations (Wu et al, 2014; Choi et al, 2020; Moriyama et al, 2018). In an earlier study, we found that SFTSV NSs binding to LSm14a disrupted LSm14a-RIG-I interaction, thus blocking antiviral IFN-β response (Zhang et al, 2021). In the current study, we identified NSs as a negative regulator of Sirt6 as a deacetyltransferase of YTHDF1. The binding of NSs with Sirt6 and NSs with YTHDF1 affected the formation of the Sirt6-YTHDF1 complexes, thus promoting the degradation of the lactylated YTHDF1 (Fig. 5G). This NSs-mediated mechanism against the host's m6A modification of viral RNAs and YTHDF1-mediated degradation targeting m6A-RNAs reflects intricate interaction between invading pathogen and host's counteract.

To our best knowledge, the lactylation of YTHDF1 and its roles in SFTSV replication is the first such a report, that SFTSV exploited the lactylation as a specific host component that targets viral RNAs for degradation, thus promoting viral replication. However, the possibility that other writers or erasers also exist could not be ruled out and need further investigation.

In conclusion, this study demonstrated that SFTSV infection triggered m6A epigenetic modification of viral RNAs, which were targeted by the host's YTHDF1-mediated degradation. In counteract, the virus-induced lactylation of YTHDF1 for degradation, thus facilitating its replication. Mechanistically, SFTSV virulent factor, NSs, through competitive binding to Sirt6, inhibiting the formation of Sirt6-YTHDF1 complex and resulting in the degradation of YTHDF1, as a counteract to the host's degradation of m6A-modified viral RNAs. Our study revealed a novel mechanism of m6A-mediated viral RNA degradation as an efficient host defense and the viral virulence factor-mediated lactylation of a key m6A reader for disruption of this defense, which will provide insightful knowledge to the therapeutic development against SFTSV infection.

## Methods

### Reagents and tools table

| Reagent/resource | Reference or source | Identifier or catalog number |
| --- | --- | --- |
| **Experimental models** | | |
| HeLa cells | ATCC | Cat #BFN60700111 |
| HUVEC cells | ATCC | Cat #BFN607200285 |
| HEK-293T cells | ATCC | Cat #BFN60700191 |
| **Recombinant DNA** | | |
| PECMV-YTHDF1 | This study | |
| PECMV-NSs | This study | |
| PECMV-NP | This study | |
| PECMV-Gn | This study | |
| pcDH-Myc-ESCO1 | Provided by Dr. Chun Lu's Laboratory at Nanjing Medical University | |
| pcDH-Myc-ESCO2 | Provided by Dr. Chun Lu's Laboratory at Nanjing Medical University | |
| pcDH-Myc-ATAT1 | Provided by Dr. Chun Lu's Laboratory at Nanjing Medical University | |
| pcDH-Myc-MYST1 | Provided by Dr. Chun Lu's Laboratory at Nanjing Medical University | |
| pcDH-Sirt1 | Provided by Dr. Chun Lu's Laboratory at Nanjing Medical University | |
| pcDH-Sirt2 | Provided by Dr. Chun Lu's Laboratory at Nanjing Medical University | |
| pcDH-Sirt3 | Provided by Dr. Chun Lu's Laboratory at Nanjing Medical University | |
| pcDH-Sirt3 | Provided by Dr. Chun Lu's Laboratory at Nanjing Medical University | |

| Reagent/resource | Reference or source | Identifier or catalog number |
|---|---|---|
| pcDH-Sirt4 | Provided by Dr. Chun Lu's Laboratory at Nanjing Medical University | |
| pcDH-Sirt5 | Provided by Dr. Chun Lu's Laboratory at Nanjing Medical University | |
| pcDH-Sirt6 | Provided by Dr. Chun Lu's Laboratory at Nanjing Medical University | |
| pcDH-Sirt7 | Provided by Dr. Chun Lu's Laboratory at Nanjing Medical University | |
| HA-ub | This study | |
| pECMV-K517R-YTHDF1 | This study | |
| pECMV-K521R-YTHDF1 | This study | |
| pECMV-K26RYTHDF1 | This study | |
| pECMV-K240RYTHDF1 | This study | |
| pECMV-K249RYTHDF1 | This study | |
| pECMV-K517R-YTHDF1 | This study | |
| pECMV-K521R-YTHDF1 | This study | |
| **Antibodies** | | |
| Rabbit-anti-YTHDF1 | Proteintech | Cat #17479-1-AP |
| Mouse-anti-YTHDF1 | Proteintech | Cat #66745-1-Ig |
| Rabbit-antiYTHDF2 | Proteintech | Cat #24744-1-AP |
| Rabbit-antiYTHDF3 | Sigma | Cat #SAB2108258 |
| Rabbit-antiALKBH5 | Proteintech | Cat #16837-1-AP |
| Mouse-antiMETTL3 | Proteintech | Cat #15073-1-AP |
| Mouse-antiGAPDH | Proteintech | Cat #60004-1-Ig |
| Rabbit anti-Klac | PTM Biolabs | Cat #PTM-1401 |
| Rabbit anti-Kbu | PTM Biolabs | Cat #PTM-329 |
| Rabbit anti-Kac | PTM Biolabs | Cat #PTM-105 |
| Mouse anti-Kcr | PTM Biolabs | Cat #PTM-502 |
| Mouse anti-Kpr | PTM Biolabs | Cat #PTM-203 |
| Mouse anti-Khib | PTM Biolabs | Cat #PTM-802 |
| Mouse anti-SFTSV NP | Cambridge Bio | Cat #1-05-0130 |
| Rabbit anti-SFTSV NSs | ABclonal | Cat #E7914 |
| Goat anti-M800 | LI-COR Inc. | Cat #926-32210 |
| Goat anti-M680 | LI-COR Inc. | Cat #926-68070 |
| Goat anti-R800 | LI-COR Inc. | Cat #926-32211 |
| Goat anti-R680 | LI-COR Inc. | Cat #926-68071 |
| Donkey anti-M488 | Jackson ImmunoResearch Inc. | Cat #715-547-003 |

| Reagent/resource | Reference or source | Identifier or catalog number |
|---|---|---|
| Donkey anti-R488 | Jackson ImmunoResearch Inc. | Cat #711-547-003 |
| Donkey anti-M594 | Jackson ImmunoResearch Inc. | Cat #715-587-003 |
| Donkey anti-R594 | Jackson ImmunoResearch Inc. | Cat #711-587-003 |
| **Oligonucleotides and other sequence-based reagents** | | |
| Plasmids | This study | Table EV1 |
| PCR primers | This study | Table EV3 |
| siRNAs | This study | Table EV3 |
| **Chemicals, enzymes, and other reagents** | | |
| Lipo8000 | Beyotime | Cat #C0533 |
| Lipofectamine 3000 | Thermo Fisher Scientific | Cat #L3000015 |
| RIPA Lysis Buffer System | Santa Cruz | Cat #sc-364162A |
| TRIzol | Invitrogen | Cat #15596026CN |
| RNase | Beyotime | Cat #D7089 |
| RNase inhibitor | Beyotime | Cat #R0102 |
| Sodium L-lactate | Selleck | Cat #S6010 |
| Cycloheximide | Selleck | Cat #S7418 |
| Actinomycin D | Selleck | Cat #S8964 |
| Firefly Luciferase Reporter Gene Assay Kit | Beyotime | Cat #RG005 |
| TGX Stain-Free FastCast acrylamide kits | BIO-RAD | Cat #1610184 |
| **Software** | | |
| SRAMP | http://www.cuilab.cn/sramp | |
| DeepKla | http://lin-group.cn/server/DeepKla/index.html | |
| **Other** | | |

## Cell culture and virus

HeLa cells (ATCC, USA), HUVEC cells (ATCC, USA), and HEK-293T cells (ATCC, USA) were maintained in DMEM (Gibco, USA) supplemented with 10% (v/v) heat-inactivated FBS (Gibco, USA), 100 µg/mL streptomycin, 100 U/mL penicillin at 37 °C and an atmosphere of 5% $CO_2$. The SFTSV of subtype E-JS-2013-24 was provided by Jiangsu Provincial CDC, Nanjing, China. The virus was passaged in Vero E6 cells.

## Western blotting

Cell lysates were prepared in standard RIPA lysis buffer (Santa Cruz, USA) and cleared by centrifugation. Total proteins were quantified by BCA protein assay kit (Life Technologies, USA), separated on 10% SDS-polyacrylamide gel, and transferred onto PVDF membranes (Millipore, MA, USA). Proteins were detected

with respective antibodies at 4 °C overnight, followed by incubation with secondary antibody. The images were scanned and quantified by densitometric analysis on LI-COR Odyssey Infrared Imager. All antibodies used in this study are listed in Table EV2.

## RNA isolation, quantitative real-time PCR (qPCR), and sequencing analysis

TRIzol reagent (Takara, CA, USA) was used to extract total RNA from cells, and cDNAs were generated by HiScript III RT SuperMix for qPCR (+gDNA wiper) Kit (Vazyme, Nanjing, China). After reverse transcription was completed, the ABI Prism 7500 Sequence Detection System analyzed the RNA transcript levels. All samples were normalized to GAPDH or 18 s rRNA. All primers used in this study are listed in Table EV3.

## Plasmids and siRNA transfection

Plasmids and siRNA transfections were performed using the Lipofectamine 3000 transfection reagent (Thermo Fisher) or Lipo8000 (Beyotime) according to the manufacturer's instructions. In brief, 70% confluent HeLa cells or 293T cells in 12-well plates were transfected with 1 ug of plasmid or 30pmol of siRNA and 36 h later infected with SFTSV. At 36 h after infection, cells were lysed in RIPA lysis buffer (Santa Cruz, USA) on ice and collected for western blot. All plasmids and siRNAs used in this study are listed in Tables EV1 or EV3.

## RNA immunoprecipitation and quantitative real-time PCR (RIP-qPCR)

RNA immunoprecipitation was performed using the Magna RIP™ RNA Binding Protein Immunoprecipitation Kit (Millipore, MA, USA), according to the manufacturer's instructions. Briefly, antibodies conjugated protein A/G magnetic beads for 4 h. Cells were washed twice with PBS, collected, and then the pellet was resuspended in IP lysis buffer. The lysate was harvested by centrifugation at $12,000 \times g$ for 10 min and incubated with beads for 30 min. Next, immunoprecipitates were eluted and subjected to proteinase K treatment, followed by RNA extraction using TRIzol (Takara, CA, USA). The relative interaction between protein and target RNA was determined by RT-PCR and normalized to input.

## Immunoprecipitation (IP)

Protein immunoprecipitation was performed using the Pierce™ Protein A/G Magnetic Beads (Thermo Fisher, USA). Following the manufacturer's instructions, antibodies conjugated protein A/G magnetic beads for 4 h. Cells were washed twice with PBS, collected, and then the pellet was resuspended in IP lysis buffer. The lysate was harvest by centrifugation at $12,000 \times g$ for 10 min and incubated with beads for 30 min. After extensive washing, bound proteins were processed by western blotting using the corresponding antibodies.

## MeRIP-seq

MeRIP-seq was performed by Romics Technology Co., Ltd. (Shanghai, China) according to the manufacturer's instructions. Total RNA from SFTSV-infected HeLa cells was isolated, purified, fragmented, and subjected to m6A RNA immunoprecipitation. Eluted m6A-containing fragments (IP) and untreated input control fragments were converted to the final cDNA library following strand-specific library preparation using the dUTP method. The average insert size for the paired-end libraries was ~100–200 bp. Paired-end sequencing was performed on an Illumina Novaseq™ platform.

## Immunofluorescence and confocal microscopy (IF)

Cell slides were fixed with 4% paraformaldehyde for 25 min and treated with 0.02% Triton-X100 for 15 min. Then, incubated with 2% BSA (wt/vol) for an hour and indicated primary antibody overnight. Subsequently, the slides were incubated with fluorescence-labeled secondary antibody (Life, NY, USA), and nuclear staining was mounted with 4', 6-diamidino-2-phenylindole (DAPI). Finally, slides were sealed by fluorescence antiquencher reagent (Invitrogen, CA, USA). Images were taken using an Olympus confocal microscope FV3000.

## mRNA stability assay

293T cells were transfected with cDNA of Human-YTHDF1 and SFTSV NP, then treated with 5 ug/mL actinomycin D (Cat#: S8964, Selleck, TX, USA) for 0, 3, 6, 9, and 12 h. The total RNA was extracted using TRIzol reagent (Takara, CA, USA). Quantitative real-time PCR (RT-qPCR) was performed to analyze mRNA levels.

## Luciferase reporter assay

293T cells were seeded into a 24-well plate and transfected with luciferase reporter vectors. After 36 h, the cells were lysed and tested using a dual-luciferase reporter assay kit (Beyotime, Shanghai, China) according to the manufacturer's instructions. Independent experiments were conducted in triplicate.

## Protein stability assay

HeLa and HUVEC cells were treated with or without 10 mM sodium ʟ-lactate 12 h and treated with deacetylase inhibitors (Beyotime, Shanghai, China) for 4 h and then treated with 100 μg/mL CHX for 0 h, 2 h, and 4 h before harvesting. The WCLs were analyzed by western blot.

## Lactic acid assay

HeLa and HUVEC cells were maintained in Pyruvic acid-free DMEM with 10% FBS, and infected with or without SFTSV. Then, cell supernatants and whole cell lysates were collected and tested by lactic Acid assay kit (Jian Cheng bio, Nanjing, China) per manufacturer's protocols. The lactic acid dehydrogenated to pyruvic acid and the absorbance was tested in 530 nm.

## Statistical analysis

All quantitative data are presented as means ± SEM. Significance between specific data sets is described in the respective figure legend and was performed using ANOVA or independent sample *t* test in GraphPad Prism 9 software.

## Text refinement

After drafting our manuscript, we only employed OpenAI's ChatGPT language model to polish and edit the text, aiming to enhance clarity and expression. Throughout the editing process, we adhered to principles of academic integrity, preserving the original authors' ideas and viewpoints.

# Data availability

The MeRIP-seq data have been deposited in NCBI's Gene Expression Omnibus and are accessible through GEO Series accession number GSE256528.

The source data of this paper are collected in the following database record: biostudies:S-SCDT-10_1038-S44319-024-00310-7.

# Peer review information

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

## Acknowledgements

This work was supported by grants from the National Natural Science Foundation of China (31970149 and U22A20335 to WZW).

## Author contributions

**Bingxin Liu**: Conceptualization; Data curation; Formal analysis; Investigation; Writing—original draft. **Xiaoyan Tian**: Data curation; Software. **Linrun Li**: Data curation. **Rui Zhang**: Data curation. **Jing Wu**: Data curation; Software. **Na Jiang**: Software. **Meng Yuan**: Visualization. **Deyan Chen**: Visualization. **Airong Su**: Visualization. **Shijie Xu**: Software. **Zhiwei Wu**: Conceptualization; Supervision; Funding acquisition; Project administration; Writing—review and editing.

Source data underlying figure panels in this paper may have individual authorship assigned. Where available, figure panel/source data authorship is listed in the following database record: biostudies:S-SCDT-10_1038-S44319-024-00310-7.

## Disclosure and competing interests statement

The authors declare no competing interests.

# Expanded View Figures

**Figure EV1. The regulatory relationship between SFTSV infection and m6A modification-related enzymes.** ▶

(A) Immunofluorescence assay was performed to measure the distribution of METTL3 (Green) in HeLa cells after SFTSV infection. Cell nuclei were stained with DAPI. Scale bar = 10 μm. (B) Immunofluorescence assay was performed to measure the distribution of METTL14 (Green) in HeLa cells after SFTSV infection. Cell nuclei were stained with DAPI. Scale bar = 10 μm. (C) Immunofluorescence assay was performed to measure the distribution of ALKBH5 (Red) in HeLa cells after SFTSV infection. Cell nuclei were stained with DAPI. Scale bar = 10 μm. (D) HeLa cells were transfected with siMETTL3 or siNC for 36 h, and then infected with SFTSV for 36 h. The expression levels were detected by western blot. GAPDH was used as a loading control. (E) HeLa cells were transfected with siALKBH5 or siNC for 36 h, and then infected with SFTSV for 36 h. The expression levels were detected by western blot. GAPDH was used as a loading control. (F) HeLa cells were transfected with PECMV-METTL3 plasmid for 36 h, and then infected with SFTSV for 36 h. The expression levels were detected by western blot. GAPDH was used as a loading control. (G) HeLa cells were transfected with PECMV-ALKBH5 plasmid for 36 h, and then infected with SFTSV for 36 h. The expression levels were detected by western blot. GAPDH was used as a loading control.

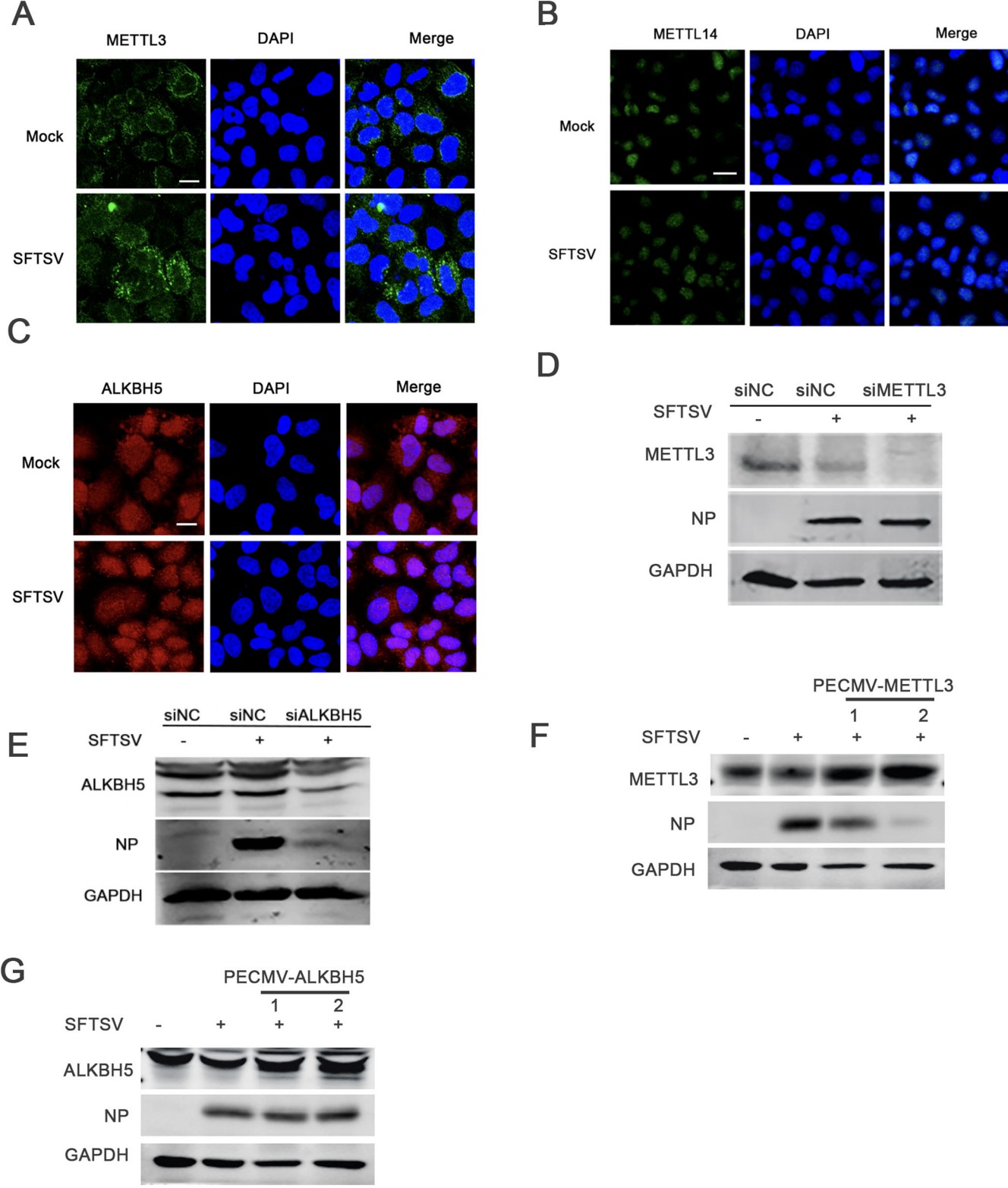

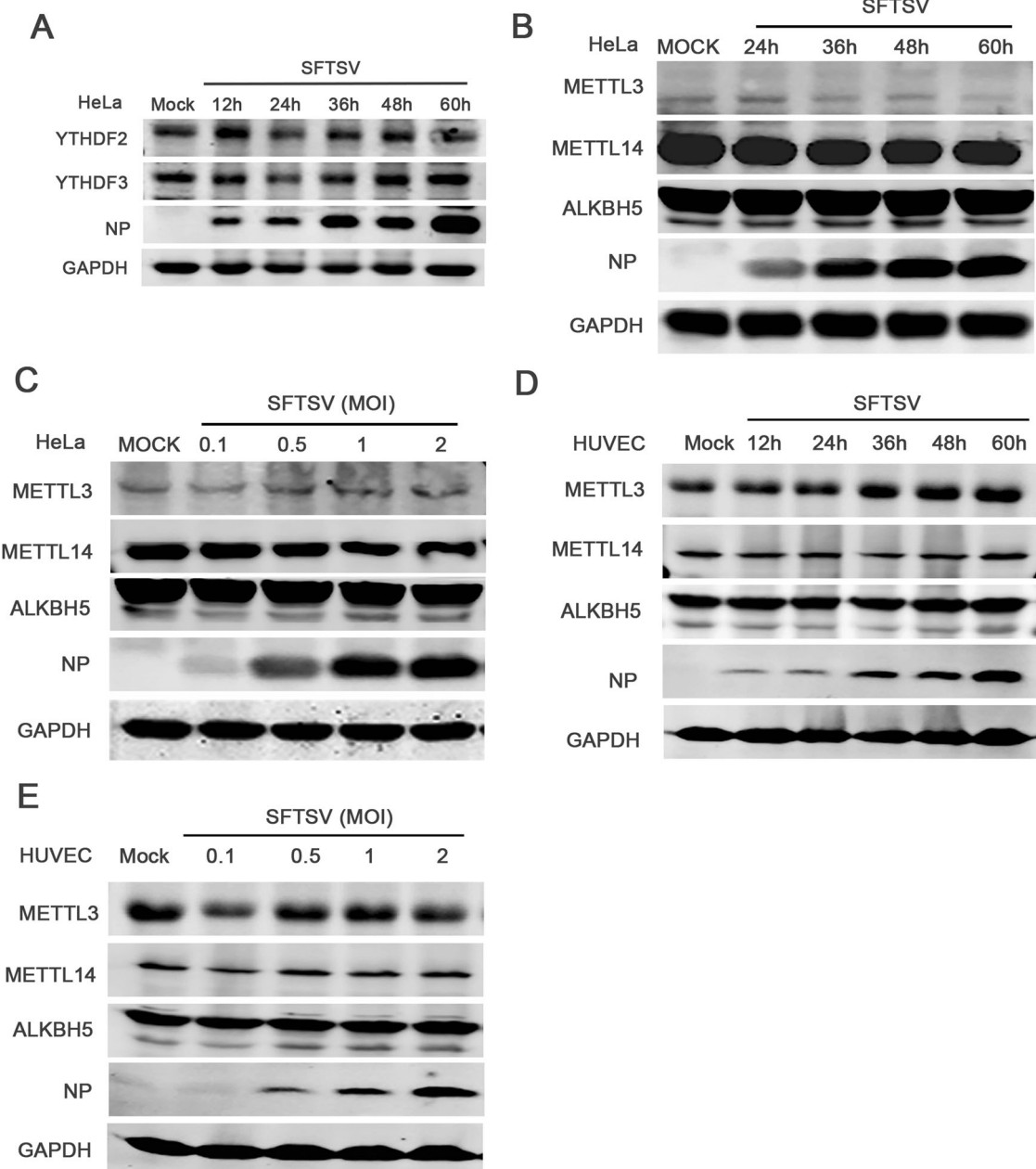

**Figure EV2. The expression of m6A-related methyltransferases, demethylases, and reader proteins in HeLa and HUVEC cells after SFTSV infection.**

(A) HeLa cells were infected with SFTSV (MOI = 1) in a time-dependent manner, and the expression level of YTHDF2 and YTHDF3 protein were detected by western blot. GAPDH was used as a loading control. (B) HeLa cells were infected with SFTSV in a time-dependent manner. The expression of ALKBH5, METTL3, and METTL14 were detected by western blot. (C) HeLa cells were infected with SFTSV in a dose-dependent manner. The expression of ALKBH5, METTL3, and METTL14 were detected by western blot. (D) HUVEC cells were infected with SFTSV in a time-dependent manner. The expression of ALKBH5, METTL3, and METTL14 were detected by western blot. (E) HUVEC cells were infected with SFTSV in a dose-dependent manner. The expression of ALKBH5, METTL3, and METTL14 were detected by western blot.

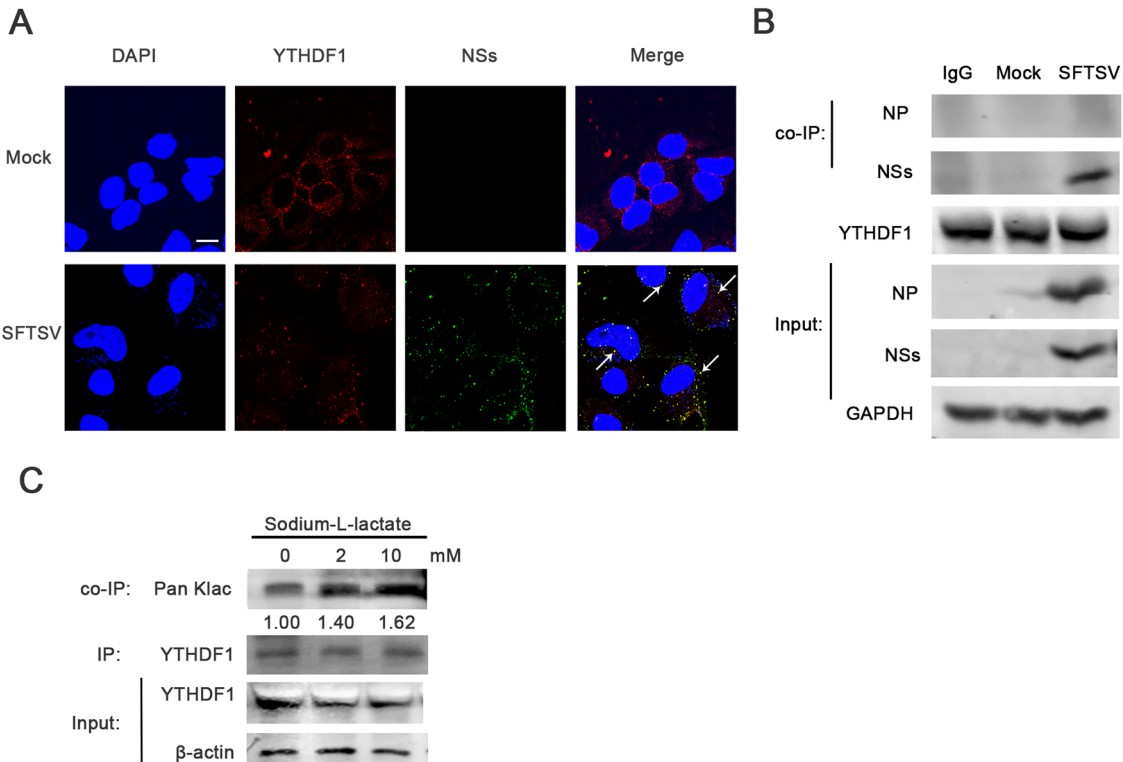

**Figure EV3.** The virulence factor NSs of SFTSV co-localized with YTHDF1.

Exogenously added Soudium-L-lactate could increase the lactylation modification on YTHDF1. (A) Immunofluorescence assay was used to measure the expression level of YTHDF1 (red) and the colocalization of YTHDF1 with NSs (green) in HeLa cells after SFTSV infection. Cell nucleus were stained with DAPI. Scale bar = 10 µm. (B) The interaction between endogenous YTHDF1 and NSs was detected by endogenous IP using a YTHDF1-specific antibody in uninfected or SFTSV-infected (MOI = 1) HeLa cells. Non-specific IgG antibody was used as negative control. (C) HeLa cells were treated with 2 mM or 10 mM Sodium L-Lactate for 12 h, and the whole cell lysates were collected by co-IP using YTHDF1-specific antibody. The lactylation level were detected by western blot using anti-Klac antibody.

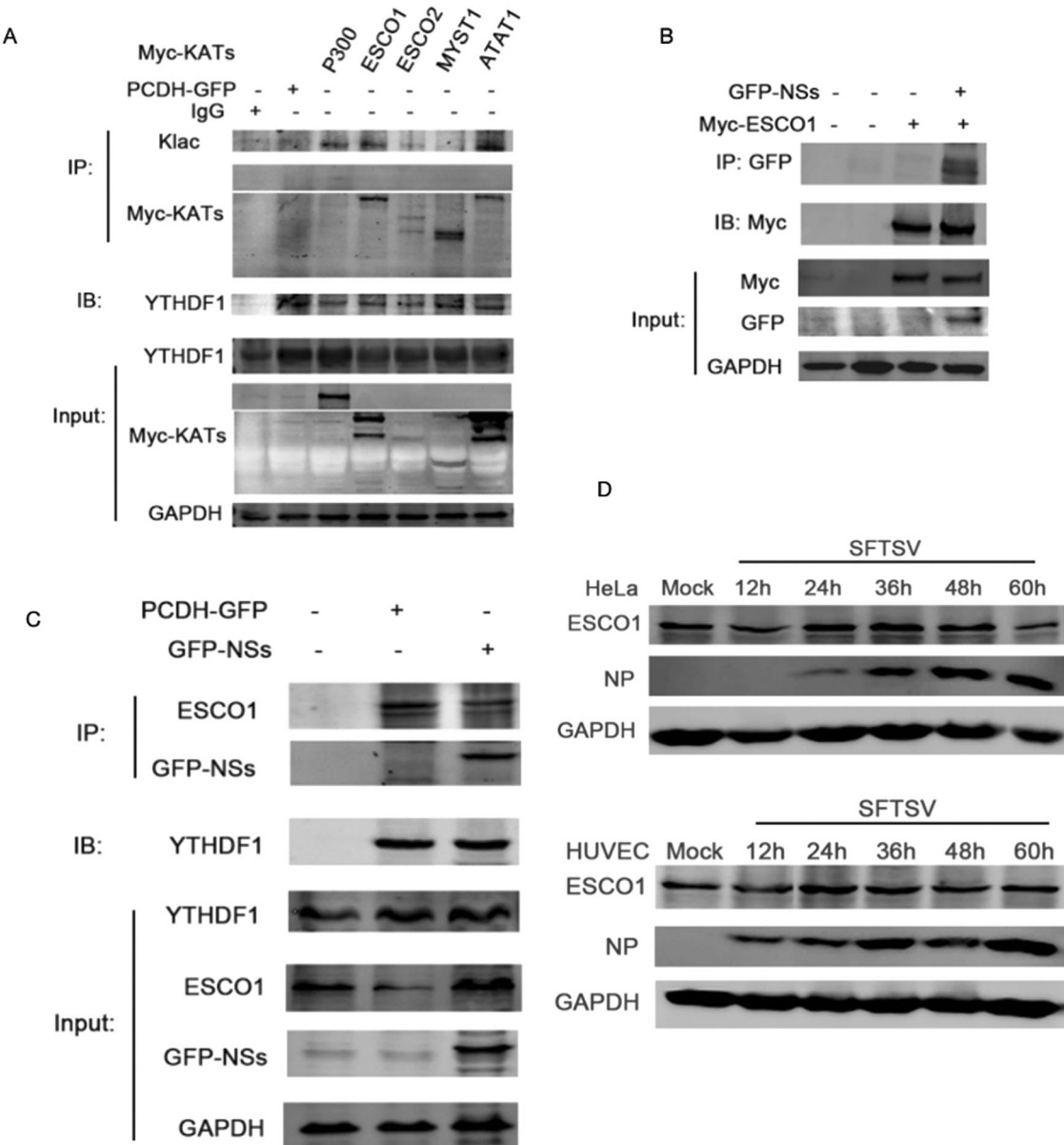

**Figure EV4. Overexpression of SFTSV NSs increased the binding affinity of YTHDF1 and ESCO1.**

(A) The interaction between endogenous YTHDF1 and Myc-P300, Myc-ESCO1, Myc-ESCO2, Myc-MYST1, and Myc-ATAT1 was detected by endogenous IP using a YTHDF1-specific antibody in HEK-293T cells. The lactylation level were detected by anti-Klac antibody. (B) The interaction between GFP-NSs and Myc-ESCO1 was detected by IP using a Myc-Specific antibody in HEK-293T cells. (C) IP was performed to measure the lactylation changes of YTHDF1 and the interaction between endogenous YTHDF1 and ESCO1 after overexpression of GFP-NSs by using YTHDF1-Specific antibody in HEK-293T cells. (D) The expression of ESCO1 protein was detected in HeLa and HUVEC cells in a time-dependent manner by western blot. GAPDH was used as a loading control.

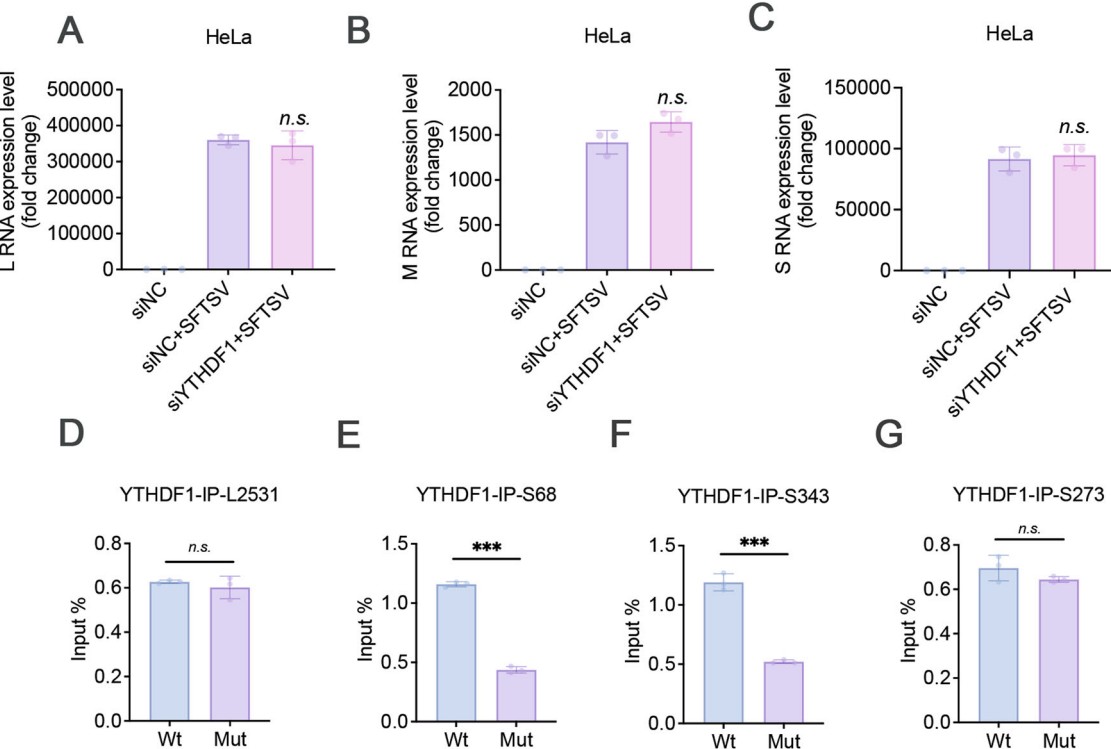

**Figure EV5.   The impact of knocking down YTHDF1 on SFTSV RNAs.**

The effect of point mutations at the m6A sites on the binding of S RNA to YTHDF1. (A–C) HeLa cells were transfected with siNC (Negative control) or siYTHDF1 for 24 h, and then infected with SFTSV (MOI = 1). The expression level of L, M, and S RNA of SFTSV were detected at 36 h by qPCR. The fold enrichment was determined by calculating the 2-Δct of the sample relative to the GAPDH. The results are represented as the means ± SD of $n = 3$ biological replicates. Statistical significance was determined by a two-sided Student's $t$ test (n.s. = 0.7023, 0.0872, 0.5662). (D-G) HEK-293T cells were transfected with Wild-type plasmids or site-mutant plasmids for 24 h. YTHDF1-IP-qRT-PCR were performed to collect m6A-modified RNAs. The fold enrichment was determined by calculating the 2-Δct of the sample relative to input. The results are represented as the mean ± SD of $n = 3$ biological replicates. Statistical significance was determined by a two-sided Student's $t$ test (L2531: n.s. = 0.4370, S68: ***$P < 0.0001$, S343: ***$P < 0.0001$, S273: n.s. = 0.2055).

