## [Peer Review File · EMBO Reports]

SFTS Virus Induces Lactylation of m6A Reader Protein YTHDF1 to Facilitate viral Replication

Bingxin Liu, Xiaoyan Tian, Linrun Li, Rui Zhang, Jing Wu, Na Jiang, Meng Yuan, Deyan Chen, Airong Su, Shijie Xu, and Zhiwei Wu

Corresponding author(s): Zhiwei Wu (wzhw@nju.edu.cn)

Review Timeline:

Submission Date:	17th Jun 24
Editorial Decision:	25th Jul 24
Revision Received:	16th Sep 24
Editorial Decision:	10th Oct 24
Revision Received:	12th Oct 24
Accepted:	22nd Oct 24

Editor: Esther Schnapp

Transaction Report:

Dear Dr. Wu,

Thank you for the submission of your manuscript to EMBO reports. We have now received the comments from 2 referees, which are pasted below. I could only secure 2 referees for your ms, despite contacting many people, but the two reports we have are good. I also asked referee 2 for cross-comments that are pasted below too.

As you will see, the referees acknowledge that the findings are potentially interesting. However, they also have significant concerns regarding m6A methylation of SFTSV RNA, and both referees agree that a validation of m6A modification of SFTSV viral RNAs using a single site-resolution validation experiment needs to be provided. I think all other concerns also need to be addressed but please let me know in case you disagree, and we can discuss the exact revision requirements further, also in a video chat, if you like.

I would thus like to invite you to revise your manuscript with the understanding that the referee concerns must be fully addressed and their suggestions taken on board. Please address all referee concerns in a complete point-by-point response. Acceptance of the manuscript will depend on a positive outcome of a second round of review. It is EMBO reports policy to allow a single round of major revision only and acceptance or rejection of the manuscript will therefore depend on the completeness of your responses included in the next, final version of the manuscript.

We realize that it is difficult to revise to a specific deadline. In the interest of protecting the conceptual advance provided by the work, we recommend a revision within 3 months (25th Oct 2024). Please discuss the revision progress ahead of this time with the editor if you require more time to complete the revisions.

- 1) A data availability section providing access to data deposited in public databases is missing. If you have not deposited any data, please add a sentence to the data availability section that explains that.
- 2) Your manuscript contains statistics and error bars based on $n=2$. Please use scatter blots in these cases. No statistics should be calculated if $n=2$.

5) a complete author checklist, which you can download from our author guidelines

<<https://www.embopress.org/page/journal/14693178/authorguide>>. Please insert information in the checklist that is also reflected in the manuscript. The completed author checklist will also be part of the RPF.

6) Please note that all corresponding authors are required to supply an ORCID ID for their name upon submission of a revised manuscript (<<https://orcid.org/>>). Please find instructions on how to link your ORCID ID to your account in our manuscript tracking system in our Author guidelines

<<https://www.embopress.org/page/journal/14693178/authorguide#authorshipguidelines>>

10) Regarding data quantification (see Figure Legends:

<https://www.embopress.org/page/journal/14693178/authorguide#figureformat>)

- the name of the statistical test used to generate error bars and P values,

- the number (n) of independent experiments (please specify technical or biological replicates) underlying each data point,

- the nature of the bars and error bars (s.d., s.e.m.),

- If the data are obtained from n Program fragment delivered error ``Can't locate object method "less" via package "than" (perhaps you forgot to load "than"?) at //ejpvfs23/sites23b/embor_www/letters/embor_decision_revise_and_review.txt line 56.' 2, use scatter blots showing the individual data points.

12) All Materials and Methods need to be described in the main text using our 'Structured Methods' format, which is required for all research articles. According to this format, the Methods section includes a Reagents and Tools Table (listing key reagents, experimental models, software and relevant equipment and including their sources and relevant identifiers) followed by a Methods and Protocols section describing the methods using a step-by-step protocol format. The aim is to facilitate adoption of the methodologies across labs. More information on how to adhere to this format as well as a downloadable template (.docx) for the Reagents and Tools Table can be found in our author guidelines:

An example of a Method paper with Structured Methods can be found here: <https://www.embopress.org/doi/full/10.1038/s44320->

I look forward to seeing a revised form of your manuscript when it is ready.

Referee #1:

SFTSV is an emerging negative-sense RNA virus. The manuscript aims to determine the relevance of m6A within the SFTSV genome. They identify m6A modifications on all three RNA segments of SFTSV. They suggest that the m6A reader binds the m6A modified RNAs, leading to their reduced stability and resulting translation. They further highlight the interesting finding that the SFTSV virulence factor NSs induces lactylation on YTHDF1, leading to its degradation and facilitating SFTSV replication.

Specific Comments:

1. The authors fail to highlight the recent finds questioning m6A modifications in RNA viruses. Questions have always remained on how do cytoplasmic RNAs viruses get m6A methylated when the cellular m6A writer is nuclear. This is emphasised by recent findings using comprehensive analysis using m6A-Seq and the antibody-independent SELECT and nanopore direct RNA sequencing techniques find no evidence of m6A modification in CHIKV or DENV transcripts. Therefore, further evidence by alternative methods - nanopore sequencing or SCARLET to map sites to single-site resolution on viral RNAs is required. In addition, as they have used a global approach of MeRIP-seq, so what changes in cellular RNAs are highlighted during infection. Moreover, MeRIP-seq has now been surpassed with more specific techniques which can map m6A sites at single site resolution and overcome the inconsistencies of m6A pulldowns.

2. Confirmation that YTHDF1 binding to SFTSV is abolished using site-directed mutants which abolish m6A modification at mapped specific residues. This emphasises the need in point 1 to map m6A sites at single site resolution.

3. Figure 2I. Is the colocalization of YTHDF1 and NSs in P bodies ?

4. Figure 2J - the interaction between YTHDF1 and NSs should be confirmed in the presence of RNase to determine this is direct and not due to RNA bridging.

5. Figure 5. Further evidence is required to shown that control/known cellular RNAs are reduced in their m6A status in METTL3 siRNA deleted cells.

In summary, the provides some interesting and novel observations that YTHDF1 is targeted for degradation by lactylation driven by SFTSV NSs. However, further evidence and assurances are required that this is due to reader activity on the m6A modification of viral RNAs, and not a general effect on m6A reading of cellular RNAs and a downstream effect on virus replication.

Referee #2:

In this manuscript the authors show that YTHDF1 is associated with m6A-modified SFTSV RNA, which downregulates YTHDF1

protein stability, in part mediated by lysine lactylation. However, there is a lack of rationale that how/why they pinned down YTHDF1 among various m6A-associated factors and why they only focused on lysine modification, not others such as arginine methylation.

Some suggestions are provided as following:

1. Figure EV1 shows METTL3 and ALKBH5 also has an effect on SFTSV infection. Why did authors decide to study on YTHDF1? Authors need to justify how they pinpointed YTHDF1 among other m6A-associated proteins.
2. The figures are not arranged in a reader-friendly way. For example, there is no explanation for Figure 2I-J until Figure 4. It is better to move them to figure 4 or corresponding EV. Authors should rearrange the figures according to the manuscript order.
3. Authors tested several modifications of YTHDF1. However, most of these are not very common modifications. Why did authors not test more common modifications such as methylation or ubiquitination? Also, there is no explanation why they only focused on lysine modification of YTHDF1. Is there any other modification such as arginine methylation regulating YTHDF1 expression in this context?
4. Authors should clearly mark IP with which antibody and IB with which antibody in IP experiments. For example, it is confusing that IP:HA looks like IP with HA antibody but it is actually not. Also, IB:flag looks like just input, but it is not.
5. Since authors showed the effect of METTL3 and ALKBH5 on NP expression upon SFTSV, it is more cohesive to explain the YTHDF1 effect (Figure 5E-F) in Figure 2.
6. Authors identified that K517 and K521 are the lactylated sites on YTHDF1. Do K517R and K521R mutants indeed increase the protein stability? And what is the effect upon SFTSV infection?
7. In Figure 3B, why is YTHDF1 level not decreased upon SFTSV infection?
8. Authors mentioned that NSs overexpression increased the Klac level of YTHDF1 (Figure 4F), but the figure does not look like it? It looks the same.

Cross-comments from referee 2:

In my opinion, reviewer#1's opinion is fair. According to a recent report (<https://www.nature.com/articles/s41467-024-46278-9>), the current report about m6A modification of viral RNAs allow us (reviewers) to have some uncertainty on SFTSV RNA's m6A. Moreover, there is currently a lack of validation of m6A modifications in the manuscript. However, we cannot simply rule out the possibility of m6A modifications on SFTS viral RNAs. SFTSV is a negative-strand RNA virus, which means they use intranuclear replication. So, they can be modified with m6A in the nucleus. The authors need to validate m6A modification of SFTS viral RNAs using a single-resolution validation experiments (eg, SELECT, MazF-digestion, and CRISPR-editing, etc).

Dear Dr. Schnapp,

I thank you very much for your time in editing the manuscript and for giving us the opportunity to revise. I also greatly appreciate the reviewers for their meticulous reading of the manuscript and raise many excellent questions. We have carefully gone through the comments and suggestions, performed additional experiments, addressed the reviewers' concerns and revised the manuscript accordingly. The revised manuscript, which has been re-submitted along with this letter, is a much-improved version and will, I wish, satisfy the rigor of the reviewers and the high standard of the journal. The point-to-point answer to the reviewers' questions is attached at the end of this letter. All the changes are highlighted in the revised manuscript. Please do not hesitate to contact me if you or the reviewers have additional questions. I am

Truly yours

Allen Z. Wu, Ph.D.
Professor and Director
Center for Public Health Research
Medical School, Nanjing University
Professor and Associate Director
State Key Laboratory of Analytical Chemistry for Life Science
Nanjing University
22 Hankou Road, Nanjing, Jiangsu 210093. China

A note for the reviewers on the change of figure annotation: since we added an additional figure as Figure 2 in the revised manuscript, all the figures after Figure 2 of the old version will have become Figure n+1 in the newly revised manuscript.

Editor and Reviewer comments:

Referee #1:

SFTSV is an emerging negative-sense RNA virus. The manuscript aims to determine the relevance of m6A within the SFTSV genome. They identify m6A modifications on all three RNA segments of SFTSV. They suggest that the m6A reader binds the m6A modified RNAs, leading to their reduced stability and resulting translation. They further highlight the interesting finding that the SFTSV virulence factor NSs induces lactylation on YTHDF1, leading to its degradation and facilitating SFTSV replication.

Specific Comments:

1. The authors fail to highlight the recent finds questioning m6A modifications in RNA viruses. Questions have always remained on how do cytoplasmic RNAs viruses get m6A methylated when the cellular m6A writer is nuclear. This is emphasised by recent findings using comprehensive analysis using m6A-Seq and the antibody-independent SELECT and nanopore direct RNA sequencing techniques find no evidence of m6A modification in CHIKV or DENV transcripts. Therefore, further evidence by alternative methods - nanopore sequencing or SCARLET to map sites to single-site resolution on viral RNAs is required. In addition, as they have used a global approach of MeRIP-seq, so what changes in cellular RNAs are highlighted during infection. Moreover, MeRIP-seq has now been surpassed with more specific techniques which can map m6A sites at single site resolution and overcome the inconsistencies of m6A pulldowns.

Response:

I thank the reviewer for the insightful comments. Nanopore sequencing technology has its advantages and also disadvantages. Nanopore sequencing technology has the ability to generate long reads of up to 2 Mb in length and provide portability. However, the biggest drawbacks to date have been a lower throughput of sequence data and a high error rate (approximately 10%) (DOI: 10.1128/JCM.01315-19) while generating accurate genome assemblies (DOI: 10.1093/bib/bby017). Krusnauskas conclude that MeRIP-seq is more suitable for an initial m6A screening study, due to its higher coverage, whereas dRNA-seq is most useful when more in-depth analysis of m6A quantity and precise location is of interest (DOI: 10.1080/15592294.2022.2163365). Nevertheless, we fully agree with the reviewer's concerns that conducting single-base resolution sequencing is both important and necessary. However, we encountered some specific technical challenges with experimental execution: In

a second-generation sequencing used in MeRIP-seq, we used a method to remove ribosomal RNA to obtain total RNA containing viral RNAs. The current nanopore sequencing methods for collecting total RNA involve capturing poly(A)-containing RNA. Unfortunately, the SFTSV genomic RNAs do not have poly(A) tails. This means that we are unable to perform third-generation sequencing on the SFTSV genomic RNAs. However, to address the reviewer's concerns, we devised alternative approaches to validate the accuracy of MeRIP-seq and made efforts to identify some m6A-modified sites on SFTSV RNAs, as described below:

- (1) Since we were unable to perform third-generation sequencing to detect potential m6A modification sites, we predicted the highest-confidence sites for m6A modification using SRAMP and intersected these with the MeRIP-seq results, and identified a total of eight sites (Table EV4). We then mutated individual sites from A to U and conducted m6A-IP-qRT-PCR. The results were not as problematic as described in the *Nature* article. Among the eight identified sites (Figure 1K-R), only L2531 (Figure 1K) showed no significant difference in RIP before and after mutation. For the remaining seven sites, the m6A-IP results were significantly reduced after the A-to-U mutation, confirming that these sites are indeed m6A-modified on SFTSV RNAs.
- (2) To substantiate the observations that SFTSV genomes could be m6A modified, we examined the cellular localization of the m6A machinery by immunofluorescence and whether any changes occurred following SFTSV infection. We observed that both METTL3 and ALKBH5 were present in both the nucleus and cytoplasm (Figure 1A, C). Following SFTSV infection, METTL3 showed a distinct characteristic of aggregation in the cytoplasm. In contrast, METTL14 remained localized in the nucleus both before and after infection (Figure 1B). To further validate this observation, we performed nuclear-cytosol fractionation experiments and found that both METTL3 and ALKBH5 were present in the cytoplasm before and after SFTSV infection. Following SFTSV infection, there was a slight decrease of METTL3 in the nucleus, accompanied by a corresponding slight increase in the cytoplasm, suggesting that METTL3 and ALKBH5 are present in the cytoplasm. Our observation is consistent with data from Uniprot (Q86U44) and other reports that METTL3 was distributed in both the cytoplasm and the nucleus ([10.1016/j.molcel.2016.03.021](https://doi.org/10.1016/j.molcel.2016.03.021), [10.1038/s41467-022-34209-5](https://doi.org/10.1038/s41467-022-34209-5)). Then we performed METTL3-IP-qRT-PCR and ALKBH5-IP-qRT-PCR, and demonstrated that METTL3 and ALKBH5 indeed colocalize with SFTSV RNAs (Figure 1 E-J). Above data (Figure 1A-C) have been added to the revised manuscript as Figure EV1A-C and (Figure 1D-R) have been added to the revised manuscript as Figure 2.

Figure 1. (A) Immunofluorescence assay was performed to determine the distribution of METTL3 (Green) in HeLa cells after SFTSV infection. Cell nuclei were stained with DAPI. Scale bar=10 μm . (B) Immunofluorescence assay was performed to determine the distribution of METTL14 (Green) in HeLa cells after SFTSV infection. Cell nuclei were stained with DAPI. Scale bar=10 μm . (C) Immunofluorescence assay was performed to determine the distribution of ALKBH5 (Red) in HeLa cells after SFTSV infection. Cell nuclei were stained with DAPI. Scale bar=10 μm . (D) The distribution of METTL3 and ALKBH5 were assessed by nuclear-cytosol fractionation experiment. The expression levels of METTL3 and ALKBH5 were analyzed by western blot. Lamin B was used as the loading control for the nuclear fraction, while Tubulin served as the loading control for the cytoplasmic fraction. (E-G) HeLa cells were infected with SFTSV for 24 hours. METTL3-IP-qRT-PCR was performed to analyze SFTSV RNAs. The fold enrichment was determined by calculating the $2^{-\Delta\text{ct}}$ of the sample relative to input. (H-J) HeLa cells were infected with SFTSV for 24 hours. ALKBH5-IP-qRT-PCR was performed to analyze SFTSV RNAs. The fold enrichment was determined by calculating the $2^{-\Delta\text{ct}}$ of the sample relative to input. (K-R) HEK-293T cells were transfected with wild-type plasmids or site-mutant plasmids for 24 h. m6A-IP-qRT-PCR was performed to analyze m6A-modified RNAs. The fold enrichment was determined by calculating the $2^{-\Delta\text{ct}}$ of the sample relative to input. The results are presented as the mean \pm SD of n=3 biological replicates. Statistical significance was determined by a two-sided Student's t-test (L3981: *** P =0.0008, L2531: $n.s.$ =0.1673, L2586: *** P <0.0001; M806: *** P =0.0006, M1860: ** P =0.0020; S68: *** P <0.0001, S273: *** P =0.0001, S343: ** P =0.0044).

2. Confirmation that YTHDF1 binding to SFTSV is abolished using site-directed mutants which abolish m6A modification at mapped specific residues. This emphasises the need in point 1 to map m6A sites at single site resolution.

Response:

The reviewer's suggestion is well taken. As in the response to Question 1, we identified seven m6A modification sites: two located in the L genomic RNA, two in the M genomic RNA, and three in the S genomic RNA. The S genomic RNA encodes the virulence factors NSs and nucleoprotein, both of which play crucial roles in the pathogenicity of SFTSV (10.1002/jmv.28371, 10.4049/jimmunol.2100148, [10.1080/15548627.2024.2393067](https://doi.org/10.1080/15548627.2024.2393067)). In this study, we focused on the S segment, and therefore selected the three m6A modification sites within the S gene for YTHDF1-qRT-PCR analysis, with L2531 serving as a negative control. The results are shown below:

- 1) In the L2531 group, as a negative control, we found that the A2531U mutation did not affect the binding of RNA to YTHDF1 (Figure 2A);
- 2) However, mutations at A68T and A343T resulted in a significant reduction in RNA binding to YTHDF1, indicating that these two sites are effective binding sites for YTHDF1 (Figure 2B, C);

- 3) Although position 273 is an m6A modification site, the mutation did not affect the binding between S273 and YTHDF1. We speculated that this position may not be a preferred binding site for YTHDF1 (Figure 2D).

Following figures have been added to the revised manuscript as Figure EV5D-G.

Figure 2. (A-D) HEK-293T cells were transfected with wild-type plasmids or site-mutant plasmids for 24 h. YTHDF1-IP-qRT-PCR were performed to analyze m6A-modified RNAs. The fold enrichment was determined by calculating the $2^{-\Delta\text{ct}}$ of the sample relative to input. The results are presented as the mean \pm SD of n=3 biological replicates. Statistical significance was determined by a two-sided Student's t-test (L2531: *n.s.*=0.4370, S68: ****P* < 0.0001, S343: ****P* < 0.0001, S273: *n.s.*=0.2055).

3. Figure 2I. Is the colocalization of YTHDF1 and NSs in P bodies?

Response:

I thank the reviewer for the insightful comments. To address the question, we performed an additional experiment to show whether YTHDF1 and NSs colocalize in P-bodies using DDX6 as a marker for P-bodies. Since the custom anti-NSs antibody has been depleted and we have clear evidence of YTHDF1 colocalizing with NSs, we examined the localization of YTHDF1 and DDX6 in cells post-infection. Through IF, we observed that although YTHDF1 aggregated following SFTSV infection, it did not colocalize with DDX6; therefore, YTHDF1 is not co-localized in P bodies. We did not do the analysis for NSs since no commercial antibody is available. The results are shown below:

Figure 3. Immunofluorescence assay was performed to measure the distribution of P bodies (Green) and YTHDF1 (Red) in HeLa cells after SFTSV infection. Cell nuclei were stained with

DAPI. Scale bar=10 μ m.

4. *Figure 2J - the interaction between YTHDF1 and NSs should be confirmed in the presence of RNase to determine this is direct and not due to RNA bridging.*

Response:

The reviewer's suggestion is well taken. Figure 2J has been renamed as Figure EV3B following the suggestion of Reviewer #2. We performed an additional experiment with the presence of RNase A (10 μ g/mL) and we found that even after adding RNase, YTHDF1 was still able to pull down NSs. The result is further confirmed by YTHDF1-co-IP experiment with overexpression of NSs plasmids as shown in Figure 4E of this study. The results are shown below:

Figure 4. (A) The interaction between endogenous YTHDF1 and NSs was detected by co-IP using a YTHDF1-specific antibody in uninfected or SFTSV-infected (MOI=1) HeLa cells. Whole cell lysates were pre-treated with RNase before being co-incubated with YTHDF1 antibodies. Non-specific IgG antibody was used as negative control.

5. *Figure 5. Further evidence is required to shown that control/known cellular RNAs are reduced in their m6A status in METTL3 siRNA deleted cells.*

Response:

The reviewer's suggestion is well taken. We performed RNA dot blot analysis of Hela cells treated with siMETTL3 and found that siMETTL3 leads to an overall reduction in intracellular m6A modifications. The results are shown below:

Figure 5. HeLa cells were transfected with siMETTL3 for 36 h and m6A modification level was detected by dot blot.

In summary, the provides some interesting and novel observations that YTHDF1 is targeted for degradation by lactylation driven by SFTSV NSs. However, further evidence and assurances are required that this is due to reader activity on the m6A modification of viral RNAs, and not a general effect on m6A reading of cellular RNAs and a downstream effect on virus replication.

Referee #2:

In this manuscript the authors show that YTHDF1 is associated with m6A-modified SFTSV RNA, which downregulates YTHDF1 protein stability, in part mediated by lysine lactylation. However, there is a lack of rationale that how/why they pinned down YTHDF1 among various m6A-associated factors and why they only focused on lysine modification, not others such as arginine methylation.

Some suggestions are provided as following:

1. Figure EV1 shows METTL3 and ALKBH5 also has an effect on SFTSV infection. Why did authors decide to study on YTHDF1? Authors need to justify how they pinpointed YTHDF1 among other m6A-associated proteins.

Response:

I thank the reviewer for raising the question. The reasons for selecting YTHDF1 for our study instead of other m6A-related proteins are as follows:

- 1). In our research, we found that knockdown and overexpression of METTL3 and ALKBH5 indeed regulated SFTSV. However, we observed that SFTSV infection did not downregulate other m6A machinery except for YTHDF1 (Figure 6A-D, Figure 3A-D in manuscript, Figure EV 2A), prompting us to hypothesize that YTHDF1 might have a more substantial impact on SFTSV;
- 2). We hypothesized that METTL3 and ALKBH5 influence SFTSV by modulating the m6A modification peaks on SFTSV RNAs, thereby affecting YTHDF1's binding to SFTSV RNAs. In fact, METTL3 and ALKBH5 were found to play important roles in epigenetic modulated virus-induced ferroptosis. We have submitted another manuscript that explores the relationship between SFTSV-induced ferroptosis and m6A, which is under review.

Following figures have been added as Figure EV2B-E in the revised manuscript.

Figure 6. (A) HeLa cells were infected with SFTSV and analyzed at various time points. The expression of ALKBH5, METTL3, and METTL14 were detected by western blot. (B) HeLa cells were infected with SFTSV and analyzed at various dose points. The expression of ALKBH5, METTL3, and METTL14 were detected by western blot. (C) HUVEC cells were infected with SFTSV and analyzed at various time points. The expression of ALKBH5, METTL3, and METTL14 were detected by western blot. (D) HUVEC cells were infected with SFTSV and analyzed at various dose points. The expression of ALKBH5, METTL3, and METTL14 were detected by western blot.

2. The figures are not arranged in a reader-friendly way. For example, there is no explanation for Figure 2I-J until Figure 4. It is better to move them to figure 4 or corresponding EV. Authors should rearrange the figures according to the manuscript order.

Response:

I appreciate the reviewer's valuable comments. We have rearranged the layout of the figures in the revised manuscript to better align with the overall logic of the text. Figure 2I-J has been changed to Figure EV3A-B; Figure 6B has been changed to Figure EV5A-C; Figure 5I-J has been changed to Figure 3I-J.

3. Authors tested several modifications of YTHDF1. However, most of these are not very common modifications. Why did authors not test more common modifications such as methylation or ubiquitination? Also, there is no explanation why they only focused on lysine modification of YTHDF1. Is there any other modification such as arginine methylation regulating YTHDF1 expression in this context?

Response:

Thanks for your comments. The reasons for focusing on lysine modification instead of other modifications are as follows:

- 1) Our collaborators have a well-established experimental system specifically designed to detect novel lysine modifications in proteins, which makes it feasible to screen for the involvement of lysine modification in the degradation process of YTHDF1;
- 2) Exploring whether novel epigenetic modifications exist on YTHDF1 holds greater research potential and could open new avenues for understanding protein regulation, and lactylation of YTHDF1 is one of such.
- 3) The discovery of lactylation on YTHDF1 and its regulation by this modification is particularly exciting because SFTSV infection significantly alters cellular lactate levels. The role of lactate metabolism in regulating the host response to SFTSV infection remains largely unexplored. Although our current findings are limited in its scope, we aim to further investigate how SFTSV affects the host by influencing lactate metabolism in future studies;
- 4) Recent studies have identified other post-translational modifications on YTHDF1, such as ubiquitination (DOI: 10.1172/JCI169365), O-GlcNAcylation (10.1038/s41556-023-01258-x) and SUMOylation (10.1128/mbio.03168-23) (Page 4, line 108-112). In subsequent research, we plan to further explore the potential post-translational modification sites on YTHDF1.

4. Authors should clearly mark IP with which antibody and IB with which antibody in IP experiments. For example, it is confusing that IP:HA looks like IP with HA antibody but it is actually not. Also, IB:flag looks like just input, but it is not.

Response:

Thank you for pointing out. We have corrected these errors in the manuscript (Figure 4A, 4B, 4F, 4G, 5A-C, 5E).

5. Since authors showed the effect of METTL3 and ALKBH5 on NP expression upon SFTSV, it is more cohesive to explain the YTHDF1 effect (Figure 5E-F) in Figure 2.

Response:

The reviewer's suggestion is well taken. Since we added an additional figure before previous Figure 2, the old Figure 2 has been renamed as Figure 3, and the figures placements have been rearranged according to your suggestions. Previous Figure 5E-F have been into Figure 3I-J in the revised manuscript.

6. Authors identified that K517 and K521 are the lactylated sites on YTHDF1. Do K517R and K521R mutants indeed increase the protein stability? And what is the effect upon SFTSV infection?

Response:

The reviewer raised an excellent question. To determine whether K517R or K521R mutants increase the protein stability, we overexpressed Wt-YTHDF1, K517R-YTHDF1, and K521R-YTHDF1 plasmids in HeLa cells and assessed their half-life 36 h post-transfection. We found that, under exogenous Sodium L-lactate treatment, the half-life of YTHDF1 was extended after K to R point mutations. Additionally, overexpression experiments with Wt-YTHDF1, K517R-YTHDF1, or K521R-YTHDF1 revealed that YTHDF1 with mutations at the lactylation sites exerts some inhibitory effect on SFTSV (Figure 7D). The results are shown below:

Figure 7. (A-C) HEK-293T cells were transfected with Wt-YTHDF1, K517R-YTHDF1, or K521R-YTHDF1 for 30 h, and treated with 10mM Sodium L-Lactate for 6 h. Then the cells were treated with the protein synthesis inhibitor cycloheximide (CHX; 100µg/mL) for the indicated periods before harvesting. β-actin was used as a loading control. (D) HEK-293T cells were transfected with the indicated YTHDF1 plasmids and then infected with SFTSV for 36 h. The expression level of NP was detected by western blot.

7. In Figure 3B, why is YTHDF1 level not decreased upon SFTSV infection?

Response:

The reason is that our assay time was 24 h, at which time there was no significant downregulation of YTHDF1.

8. Authors mentioned that NSs overexpression increased the Klac level of YTHDF1 (Figure 4F), but the figure does not look like it? It looks the same.

Response:

Thank you for your comments. We think that this might be because the exposure time was too long so that the density was too high to differentiate the difference. Nevertheless, to re-confirm the results, we re-ran this batch of samples and the results were consistent with our conclusion. The new data is in Figure 5E.

Cross-comments from referee 2:

In my opinion, reviewer#1's opinion is fair. According to a recent report (<https://www.nature.com/articles/s41467-024-46278-9>), the current report about m6A modification of viral RNAs allow us (reviewers) to have some uncertainty on SFTSV RNA's m6A. Moreover, there is currently a lack of validation of m6A modifications in the manuscript. However, we cannot simply rule out the possibility of m6A modifications on SFTS viral RNAs. SFTSV is a negative-strand RNA virus, which means they use intranuclear replication. So, they can be modified with m6A in the nucleus. The authors need to validate m6A modification of SFTS viral RNAs using a single-resolution validation experiments (eg, SELECT, MazF-digestion, and CRISPR-editing, etc).

Response:

We appreciate the reviewer for raising this important question. We have performed additional experiments and answer the questions accordingly as presented as Response to the question 1 of the Reviewer #1.

Dear Dr. Wu,

Thank you for the submission of your revised manuscript. We have now received the enclosed reports from the referees who still have a few more suggestions that I would like you to address and incorporate before we can proceed with the official acceptance of your manuscript. Please co-submit a final point-by-point response to the last referee comments and cross-comments.

A few editorial requests will also need to be addressed:

- Please move the Data Availability Section (DAS) to before the Acknowledgments and the Disclosure and Competing Interest Statement in between these 2 sections.
- Please answer the question about blinding in the author checklist and correct where the DAS is found in the ms file.
- A callout for Figure EV1 is missing, please add.
- Please remove the instructions and the example table from your Research and Tools table file.

Please address the following comments on the figure legends:

- Please note that the 'white arrows' defined in the legend of Figure 5d are not visible in the figure panel.
- Please note that Figures EV2, EV3, EV4, and EV5 are mislabeled as Figures EV3, EV4, EV5, and EV6, respectively, in the manuscript. This needs to be rectified.
- Please note that the titles for Figures EV1, EV2, EV3, and EV5 are missing in the manuscript. This needs to be rectified.
- Please note that the exact p-values are not provided in the legends of Figures 2i, 2o, 6b, 6c, EV5e, and EV5f.
- Please define the annotated p-values (***) as well as the exact p-value in the legend of Figure 4c, as appropriate.
- Please indicate the statistical test used for data analysis in the legend of Figure 4c.
- Please note that information related to 'n' is missing in the legend of Figure 4c.
- Please note that the error bars are not defined in the legend of Figure 4c.
- Please note that the scale bar is missing from Figures 5d and EV3a.
- Please note that the scale bar needs to be defined for Figure EV1c.
- Please note that axis gaps are not labeled appropriately in Figure 6d.

I would like to suggest some minor changes to the abstract that needs to be written in present tense. Please let me know whether you agree with this:

Severe Fever with Thrombocytopenia Syndrome Virus (SFTSV), an emerging infectious pathogen with a high fatality rate, is an enveloped tripartite segmented single-stranded negative-sense RNA virus. SFTSV infection is characterized by suppressed host innate immunity, proinflammatory cytokine storm, failure of B cell immunity and robust viral replication. m6A modification has been shown to play a role in viral infections. However, interactions between m6A modification and SFTSV infection remain poorly understood. Through MeRIP-seq, we identify m6A modifications on SFTSV RNA. We show that YTHDF1 can bind to m6A modification sites on SFTSV, decreasing the stability of SFTSV RNA and reducing translation efficiency of SFTSV proteins. The SFTSV virulence factor NSs increases lactylation of YTHDF1 and YTHDF1 degradation, thus facilitating SFTSV replication. Our findings indicate that the SFTSV protein NSs induces lactylation to inhibit YTHDF1 as a countermeasure to host's YTHDF1-mediated degradation of m6A-marked viral mRNAs.

EMBO press papers are accompanied online by A) a short (1-2 sentences) summary of the findings and their significance, B) 2-3 bullet points highlighting key results and C) a synopsis image that is exactly 550 pixels wide and 200-600 pixels high (the height is variable). The synopsis image should provide a sketch of the major findings, like a graphical abstract. Please note that text needs to be readable at the final size. Please send us this information along with the final manuscript.

Referee #1:

The authors have improved the manuscript significantly. They provide sufficient data to address this reviewers concerns regarding m6A, by demonstrating the m6A modification of SFTSV RNAs, and confirm specific sites by mutation analysis, and how it impacts RNA stability.

Overall the manuscript now provides rigorous evidence that YTHDF1 can bind and destabilise SFTSV RNAs and is therefore targeted for degradation by lactylation driven by SFTSV NSs.

Referee #2:

While authors showed METTL3 and ALKBH5 binds to SFTSV RNAs by RIP-qPCR, I found it very hard to believe METTL3 and ALKBH5 are localized in cytoplasm. There are numerous papers (and even antibody data sheets) showing METTL3 is localized in nucleus. Same as ALKBH5, (Zheng et al., Mol Cell, 2013) clearly showed nuclear localization of GFP-ALKBH5 in HeLa cells. Also, figure 1D, western blot is too messy. Two different targets in one blot made it hard to tell which band is with the non-specific bands and thus not convincing.

Cross-comments from referee 1:

I gave benefit to the doubt here on the relocalisation of METTL3 and ALBHK5 into the cytoplasm. The IF could be improved. But lots of proteins can be redistributed during virus infection. The papers quoted are in uninfected cells. An example of METTL3 redistribution is observed in SARS infection.

<https://www.ncbi.nlm.nih.gov/pmc/articles/PMC10078285/>

Perhaps more discussion and examples of the role of viral proteins and how they could be redistributed is needed.

I do agree about reviewers point on the messy immunoblot.

We thank all the referees for their meticulous reading of our responses and rigorous standard on the experimental evidence. We have addressed all the comments as detailed below.

(The comments of the editor and reviewers are highlighted in blue, which are followed by our responses.)

A few editorial requests will also need to be addressed:

- Please move the Data Availability Section (DAS) to before the Acknowledgments and the Disclosure and Competing Interest Statement in between these 2 sections.

Response: Thank you for your helpful suggestion. We have moved the DAS in between these 2 sections.

- Please answer the question about blinding in the author checklist and correct where the DAS is found in the ms file.

Response: Thank you for your suggestion. We did not use any blinding so we chose *Not applicable* as response.

- A callout for Figure EV1 is missing, please add.

Response: Sorry for the oversight and we have corrected this mistake.

- Please remove the instructions and the example table from your Research and Tools table file.

Response: We have removed the instruction and the example table as suggested.

Please address the following comments on the figure legends:

- Please note that the 'white arrows' defined in the legend of Figure 5d are not visible in the figure panel.

Response: Thank you for pointing out. We have added the white arrows in Figure 5d.

- Please note that Figures EV2, EV3, EV4, and EV5 are mislabeled as Figures EV3, EV4, EV5, and EV6, respectively, in the manuscript. This needs to be rectified.

Response: Apology for the oversight. We have rectified these mistakes.

- Please note that the titles for Figures EV1, EV2, EV3, and EV5 are missing in the manuscript. This needs to be rectified.

Response: Thank you for your suggestion. We have added the titles for Figure EV1, EV2, EV3, and EV5.

- Please note that the exact p-values are not provided in the legends of Figures 2i, 2o, 6b, 6c, EV5e, and EV5f.

Response: Thank you for pointing. The *p-values* of these figures are smaller than 0.0001; therefore, we are unable to provide specific numerical values.

*- Please define the annotated p-values (***) as well as the exact p-value in the legend of Figure 4c, as appropriate.*

Response: Thank you for your suggestion. We have added the *p-value* of Figure 4C, which is $P < 0.0001$.

- Please indicate the statistical test used for data analysis in the legend of Figure 4c.

Response: We have added respective information as suggested.

- Please note that information related to 'n' is missing in the legend of Figure 4c.

Response: Thank you for your suggestion. We have corrected this mistake.

- Please note that the error bars are not defined in the legend of Figure 4c.

Response: We have corrected this mistake.

- Please note that the scale bar is missing from Figures 5d and EV3a.

Response: We have added the scale bars for Figure 5D and EV3A.

- Please note that the scale bar needs to be defined for Figure EV1c.

Response: Thank you for your suggestion. We have defined the scale bar for Figure EV1c.

- Please note that axis gaps are not labeled appropriately in Figure 6d.

Response: Thank you for your suggestion. We have relabeled the axis gaps in Figure 6d.

I would like to suggest some minor changes to the abstract that needs to be written in present tense. Please let me know whether you agree with this:

Severe Fever with Thrombocytopenia Syndrome Virus (SFTSV), an emerging infectious pathogen with a high fatality rate, is an enveloped tripartite segmented single-stranded negative-sense RNA virus. SFTSV infection is characterized by suppressed host innate

immunity, proinflammatory cytokine storm, failure of B cell immunity and robust viral replication. m6A modification has been shown to play a role in viral infections. However, interactions between m6A modification and SFTSV infection remain poorly understood. Through MeRIP-seq, we identify m6A modifications on SFTSV RNA. We show that YTHDF1 can bind to m6A modification sites on SFTSV, decreasing the stability of SFTSV RNA and reducing translation efficiency of SFTSV proteins. The SFTSV virulence factor NSs increases lactylation of YTHDF1 and YTHDF1 degradation, thus facilitating SFTSV replication. Our findings indicate that the SFTSV protein NSs induces lactylation to inhibit YTHDF1 as a countermeasure to host's YTHDF1-mediated degradation of m6A-marked viral mRNAs.

Response: We agree with your changes and have incorporated these changes in the manuscript.

Referee #1:

The authors have improved the manuscript significantly. They provide sufficient data to address this reviewer's concerns regarding m6A, by demonstrating the m6A modification of SFTSV RNAs, and confirm specific sites by mutation analysis, and how it impacts RNA stability.

Overall the manuscript now provides rigorous evidence that YTHDF1 can bind and destabilise SFTSV RNAs and is therefore targeted for degradation by lactylation driven by SFTSV NSs.

We greatly appreciate the reviewer#1's acceptance of our responses to the questions.

Referee #2:

While authors showed METTL3 and ALKBH5 binds to SFTSV RNAs by RIP-qPCR, I found it very hard to believe METTL3 and ALKBH5 are localized in cytoplasm. There are numerous papers (and even antibody data sheets) showing METTL3 is localized in nucleus. Same as ALKBH5, (Zheng et al., Mol Cell, 2013) clearly showed nuclear localization of GFP-ALKBH5 in HeLa cells. Also, figure 1D, western blot is too messy. Two different targets in one blot made it hard to tell which band is with the non-specific bands and thus not convincing.

Response: We appreciate the reviewer's meticulous analysis of experimental data and rigorous standard on conclusions. Our experimental evidence as provided by both immunofluorescence staining and fractionation study showed that both METTL3 and ALKBH5 are localized to cytoplasm and nucleus. We sincerely apologize for the confusion caused by the presentation of Figure 1D (Figure 2A in the manuscript). In fact, there are no non-specific bands in Figure 1D. The presence of multiple bands is a characteristic of ALKBH5. However, to make the image clearer, we cropped the original image (The modified image is in Figure 2A of the manuscript.).

Regarding the article (Zheng et al., Mol Cell, 2013) that used GFP-ALKBH5 over-expression system to analyze the subcellular localization of the protein. We think it may not be as the same as the endogenous ALKBH5 expression (Red arrows). In fact, close examination of the overexpression image reveals low level ALKBH5 fluorescence in the cytoplasm which may be dampened by the strong nuclear fluorescent intensity.

(Figure 3B of Zheng et al., Mol Cell, 2013)

In addition, our observation is also supported by vast quantity of evidence as detailed below:

1. Firstly, we reviewed recent articles on the localization of ALKBH5 and METTL3. We have compiled relevant immunofluorescence data supporting the localization of ALKBH5 in the cytoplasm (Figure A) (DOI: 10.1186/s13048-024-01394-4), as well as nuclear-cytoplasmic fractionation experiments for METTL3 in the context of SARS-CoV-2 (Figure B) (DOI:

10.1016/j.celrep.2021.109091), and immunofluorescence data relating to HCoV-OC43 infection and METTL3 (Figure C) (DOI: 10.1101/gad.348320.121). All of these data align with our conclusion that both ALKBH5 and METTL3 can localize in the cytoplasm.

We believe that although many previous studies emphasized that m6A modification enzymes are located in the nucleus, growing evidence suggests that both ALKBH5 and METTL3 are also present in the cytoplasm as research progresses.

2. Secondly, we reviewed the subcellular localization information for ALKBH5 and METTL3 in the *Human Protein Atlas* database. The results showed that both ALKBH5 and METTL3 can localize in the cytoplasm (Figure D, E).

3. Finally, we reviewed the immunofluorescence localization data for ALKBH5 (Figure F, ZRB1053) and METTL3 (Figure G, HPA079662) provided by Sigma Aldrich. We found that both proteins are capable of localizing in the cytoplasm.

D

THE HUMAN PROTEIN ATLAS SECTIONS ABOUT NEWS LEARN DATA HELP

Search result (1 gene): **ALKBH5**

SUMMARY

SUBCELLULAR RNA EXPRESSION **HUMAN CELLS**

ANTIBODIES AND VALIDATION

Dictionary

- Cytosol
- Golgi apparatus
- Nucleoplasm

Subcellular proteome

- Cytosol
- Golgi Apparatus
- Nucleoplasm

Toggle channels: Target protein Nucleus Intensity Microtubules ER Objects

Thumbnail	Antibody	Cell line	Cell line RNA Expression (TPM)	Location	Single-cell variation	Cell cycle dependent variation
	HPA027196	A-431	58.9	Nucleoplasm Golgi apparatus Cytosol		
	HPA027196	U-251MG	79.2	Nucleoplasm Cytosol		
	HPA027196	U2OS	53.2	Nucleoplasm Golgi apparatus Cytosol		

E

THE HUMAN PROTEIN ATLAS SECTIONS ABOUT NEWS LEARN DATA HELP

Search result (1 gene): **METTL3**

SUMMARY

SUBCELLULAR RNA EXPRESSION **HUMAN CELLS**

ANTIBODIES AND VALIDATION

Dictionary

- Cytosol
- Golgi apparatus
- Nuclear bodies
- Nucleoplasm

Subcellular proteome

- Cytosol
- Golgi Apparatus
- Nucleoplasm

Toggle channels: Target protein Nucleus Intensity Microtubules ER Objects

Thumbnail	Antibody	Cell line	Cell line RNA Expression (TPM)	Location	Single-cell variation	Cell cycle dependent variation
	HPA079602	HAP1	60.9	Nucleoplasm Nuclear bodies Cytosol		
	HPA079602	Hep-G2	74.3	Nucleoplasm Cytosol		
	HPA079602	U2OS	23.6	Nucleoplasm Golgi apparatus Cytosol		

F

ZRB1053

Immunocytochemistry

Enhanced Validation -
Recombinant Antibody
Technology
Immunofluorescent analysis of
HeLa cells was performed using a
1:100 dilution of Cat. No.
ZRB1053, Anti-ALKBH5, clone
6G7 ZooMAb[®] Rabbit Monoclonal
and visualized with a Goat Anti-
Rabbit secondary antibody
conjugated to Alexa Fluor[™] 488.
Actin filaments have been
labeled with phalloidin (Red).
Nucleus is stained with DAPI

G

HPA079662

Immunofluorescent staining of human cell line
Hep-G2 shows lc asm &
cytosol.

HPA079662 polyclonal IF human

NEW

Cross-comments from referee 1:

I gave benefit to the doubt here on the relocalisation of METTL3 and ALKBH5 into the cytoplasm. The IF could be improved. But lots of proteins can be redistributed during virus infection. The papers quoted are in uninfected cells. An example of METTL3 redistribution is observed in SARS infection.

<https://www.ncbi.nlm.nih.gov/pmc/articles/PMC10078285/>

Perhaps more discussion and examples of the role of viral proteins and how they could be redistributed is needed.

I do agree about reviewers point on the messy immunoblot.

Response: Thank you for your suggestions. We found that the article you provided also

addresses METTL3. According to the immunofluorescence data in the article, a small amount

of METTL3 is present in the cytoplasm of uninfected Vero cells, which is consistent with the conclusions from our experimental data.

Dr. Zhiwei Wu
Nanjing university
22# Hankou Road, Nanjing
Nanjing, jiangsu 210093
China

Dear Dr. Wu,

I am very pleased to accept your manuscript for publication in the next available issue of EMBO reports. Thank you for your contribution to our journal.

Yours sincerely,
